# Highly efficient and robust π-FISH rainbow for multiplexed in situ detection of diverse biomolecules

Yingfeng Tao[1,2,12], Xiaoliu Zhou[1,2,12], Leqiang Sun[1,2], Da Lin [1,2], Huaiyuan Cai[1,2], Xi Chen[1,2], Wei Zhou[1,2], Bing Yang[1,2], Zhe Hu[1], Jing Yu[3], Jing Zhang[4], Xiaoqing Yang[5], Fang Yang [6], Bang Shen [1,2,7], Wenbao Qi[8,9], Zhenfang Fu [10], Jinxia Dai [1,2] ✉ & Gang Cao [1,2,11] ✉

In the unprecedented single-cell sequencing and spatial multiomics era of biology, fluorescence in situ hybridization (FISH) technologies with higher sensitivity and robustness, especially for detecting short RNAs and other biomolecules, are greatly desired. Here, we develop the robust multiplex π-FISH rainbow method to detect diverse biomolecules (DNA, RNA, proteins, and neurotransmitters) individually or simultaneously with high efficiency. This versatile method is successfully applied to detect gene expression in different species, from microorganisms to plants and animals. Furthermore, we delineate the landscape of diverse neuron subclusters by decoding the spatial distribution of 21 marker genes via only two rounds of hybridization. Significantly, we combine π-FISH rainbow with hybridization chain reaction to develop π-FISH+ technology for short nucleic acid fragments, such as micro-RNA and prostate cancer anti-androgen therapy-resistant marker *ARV7* splicing variant in circulating tumour cells from patients. Our study provides a robust biomolecule in situ detection technology for spatial multiomics investigation and clinical diagnosis.

Cell functional diversity arises from cellular heterogeneity and various microenvironments in sophisticated and well-organized biological systems. Determining cell types and molecular properties in the tissue context is particularly indispensable to deciphering the unique function of individual cells[1-3]. The advent of large-scale single-cell RNA sequencing (scRNA-seq) has greatly facilitated the identification of cell heterogeneity in multicellular organisms[4-6]. However, it lacks the spatial location and the surrounding tissue microenvironment information of each cell cluster. FISH technologies can resolve the spatial location of nucleic acid by specific hybridization, which has generated unprecedented insights into pinpointing cell subtypes and spatial distribution relationships at the single-cell or single-molecule levels[7-10]. FISH is widely used to study gene expression and chromosome copy number variation in many species, as the availability and specificity of

[1]State Key Laboratory of Agricultural Microbiology, Hubei Hongshan Laboratory, Huazhong Agricultural University, 430070 Wuhan, China. [2]College of Veterinary Medicine, Huazhong Agricultural University, 430070 Wuhan, China. [3]Department of Blood Transfusion, Wuhan hospital of Traditional Chinese and Western Medicine, Huazhong University of Science and Technology, 430070 Wuhan, China. [4]Department of the 1st Thoracic Medical Oncology, Hubei Cancer Hospital, Tongji Medical College, Huazhong University of Science and Technology, 430070 Wuhan, China. [5]Hospital of Huazhong Agricultural University, 430070 Wuhan, China. [6]National Key Laboratory of Crop Genetic Improvement, Huazhong Agricultural University, 430070 Wuhan, China. [7]Key Laboratory of Preventive Medicine in Hubei Province, 430070 Wuhan, Hubei Province, China. [8]College of Veterinary Medicine, South China Agricultural University, 510642 Guangzhou, China. [9]African Swine Fever Regional Laboratory of China, Guangzhou, China. [10]Departments of Pathology, College of Veterinary Medicine, University of Georgia, Athens, GA 30602, USA. [11]College of Biomedicine and Health, Huazhong Agricultural University, 430070 Wuhan, China. [12]These authors contributed equally: Yingfeng Tao, Xiaoliu Zhou. ✉e-mail: jxdai@mail.hzau.edu.cn; gcao@mail.hzau.edu.cn

antibodies lag far behind the needs of the current explosive growth in biological research, especially for the detection of noncoding RNAs and splicing variants[9,11–14]. Thus, a new generation of cutting-edge FISH technology with higher sensitivity, specificity, spatial resolution, and robustness for new-generation biology research and clinical diagnosis is desperately needed.

Recently, several approaches have been developed to improve the intensity of FISH signals, such as hybridization chain reaction (HCR)[14,15], signal amplification by exchange reaction (SABER)[16], branched DNA (bDNA) amplification[17], AmpFISH[18], and hybridization-based rolling-circle amplification (RCA)[19,20]. HCR is a nucleic acid polymerization reaction that initiates the cascade reaction of hairpin oligonucleotide self-folding. In the latest version of HCR (version 3.0), the split probe can effectively suppress the background signal[14]. SABER amplifies fluorescence signals by synthesizing conjugate hybridization scaffolds in vitro[16]. bDNA amplification approach significantly increases the intensity of the fluorescence signal without increasing the fluorescent spot size[17]. RCA, a thermostatic nucleic acid amplification technique, can also generate strong signals[20]. AmpFISH probes are prepared in vitro by RCA to amplify the detection signal[18]. High-throughput FISH approaches, including MERFISH[9,21], SeqFISH+[12], and split-FISH[22], utilize combinatorial labeling and sequential imaging to visualize the expression of hundreds to thousands of genes in individual cells. These methods overcome the limitation of detecting the number of RNA species in single cells and can elucidate the comprehensive landscape of in situ gene expression profiles.

Despite the vast progress in FISH methods, new-generation FISH technologies are still in their infancy and have various unresolved challenges that need to be further overcome. (i) They require approximately 500 bp[23] or longer nucleic acid sequences for multi-plexed target probe hybridization, limiting the detection of short RNAs (e.g., microRNA), specific splicing variants and narrow genomic loci. (ii) Several excellent works using MERFISH and seqFISH+ methods have co-detected DNA, RNA, and proteins in the same sample[13,24]. The codetection of three molecules by these methods is achieved through separated hybridization, imaging, and the subsequent integrated signal analysis, which is more time-consuming and challenging, especially for the alignment of 3D images. Thus, a more efficient and accurate one-step simultaneous triple detection method for DNA, RNA, and proteins is highly desired. (iii) It is still challenging to achieve high signal intensity and efficiency while maintaining low background noise. For example, although RCA can increase the intensity of the detection signals, achieving high enzymatic reaction efficiency in complex biological tissues remains challenging[25,26]. For the methods without amplification, enhancement of imaging sensitivity is required to visualize the weak signals but, in the meantime, increase background noise.

To solve the current challenges of FISH technologies, we develop a highly efficient multiplex π-FISH rainbow method with high-intensity signals and low background noise. This robust method has been applied in animals, plants, and pathogenic microorganisms in frozen, paraffin, and whole-mount samples. Moreover, π-FISH rainbow can simultaneously detect DNA, RNA, and proteins at the single-molecule level. Of note, we further combine π-FISH rainbow with HCR to develop π-FISH+ technology to overcome the detection limitation of short sequences. This technology can identify the androgen receptor splice variant 7 (ARV7) in circulating tumor cells (CTCs) for therapy resistance diagnosis in prostate cancer as well as visualize microRNA and noncoding RNA.

## Results

### Design and validation of π-FISH rainbow

To increase FISH efficiency, stability, sensitivity, and specificity, we developed the π-FISH rainbow method, in which the primary π-FISH target probes contain 2–4 complementary base pairs in the middle region (step i) (Fig. 1a). This design is favorable for split probes to form a π-shaped bond to increase stability during hybridization and washing, which can ultimately improve efficiency and specificity. Next, secondary U-shaped (step ii) and tertiary (step iii) U-shaped amplification probes were utilized to amplify the signal. Finally, a fluorescence signal probe was used to visualize the signal (step iv). Details of the π-FISH rainbow workflow and the probe sequences are shown in Supplementary Fig. 1 and Supplementary Data 1, respectively. Notably, by combining different fluorescence signal probes (π-FISH rainbow), we could realize multiplexed target detection in one round.

First, we compared the hybridization efficiency of π target probes (with complementary base pairs) with traditional split probes (without complementary base pairs). Our data demonstrated that signal spots detected by the π target probes were more than that of the traditional split probes when one or five pairs of π target probes were used (Fig. 1b, c and Supplementary Fig. 2a, b), suggesting that the π target probe design can increase hybridization efficiency. We found that 2–4 complementary base pairs achieved almost the highest hybridization efficiency while maintaining low background noise (Fig. 1d and Supplementary Fig. 2c, d). Next, we optimized the number of π target probes per gene and observed that 10–15 probes yielded the peak signal spots (Supplementary Fig. 2e, f). While fluorescent spots with more target probes had significantly higher signal intensities, the widths of the spots were comparable across different numbers of target probes (Supplementary Fig. 2g, h). For the amplification probes, the signal intensity produced by hybridization using our U-shaped bilateral amplification probe was higher than that produced by the traditional L-shaped unilateral amplification probe (Supplementary Fig. 2i). Meanwhile, the specificity of π target probes and amplification probes (secondary and tertiary probes) of π-FISH rainbow was validated as indicated by the absence of specific fluorescent spots in the controls (unilateral target probes, no target probes, and no amplification probes) (Supplementary Fig. 2j).

Furthermore, we tested the multiplexed target detection efficiency of π-FISH rainbow. In theory, the combination of four fluorescence signal probes can generate 15 ($C_4^1 + C_4^2 + C_4^3 + C_4^4$) different signal codes and differentiate 15 genes in one round (Supplementary Fig. 3a). To generate consistent luminance for each fluorescence channel signal during the coding of merged fluorescence signals, the number of π target probes per mRNA increases based on the number of fluorescent colors. Our data demonstrated that the overlapping ratio of two and three fluorescence signals during in situ detection in HeLa cells was 99.14% and 99.06%, respectively (Supplementary Fig. 3b, c), indicating high efficiency and accuracy of π-FISH rainbow for multiplexed detection and decoding. The detection efficiencies of Gad1 and Vip genes by multichannel decoding were 99.03% and 99.10% of single-channel detection, respectively (Supplementary Fig. 3d–h). Thus, the combination of different signal probes for multiplexed target detection of π-FISH rainbow is highly reliable.

To thoroughly assess the noise level of π-FISH rainbow, we performed in situ detection of ACTB, PPIA, and B2M genes in HeLa cells, and detection of Ctgf, Penk, and Sst genes in mouse brain tissues using different controls, including unilateral target probes, no target probes, RNase treatment, and bacterial dapB probes (Supplementary Fig. 4a–d). Our data showed negligible signal spots in the negative control groups (Supplementary Fig. 4e–h). To verify the false-positive rate of π-FISH rainbow for single or multiplexed detection, we introduced the off-target probe, bacterial dapB probes, and unilaterally specific probes as additional negative controls. The false-positive rate of π-FISH rainbow was less than 0.51% (Supplementary Fig. 5).

To further analyze the efficiency of π-FISH rainbow, we compared π-FISH rainbow with HCR, smFISH, and smFISH-FL (FL, full-length transcripts). Similar numbers and sizes of probes targeting the ACTB mRNA in HeLa cells were used for all techniques, except smFISH-FL (where the whole transcript was covered with probes) (Fig. 1e).

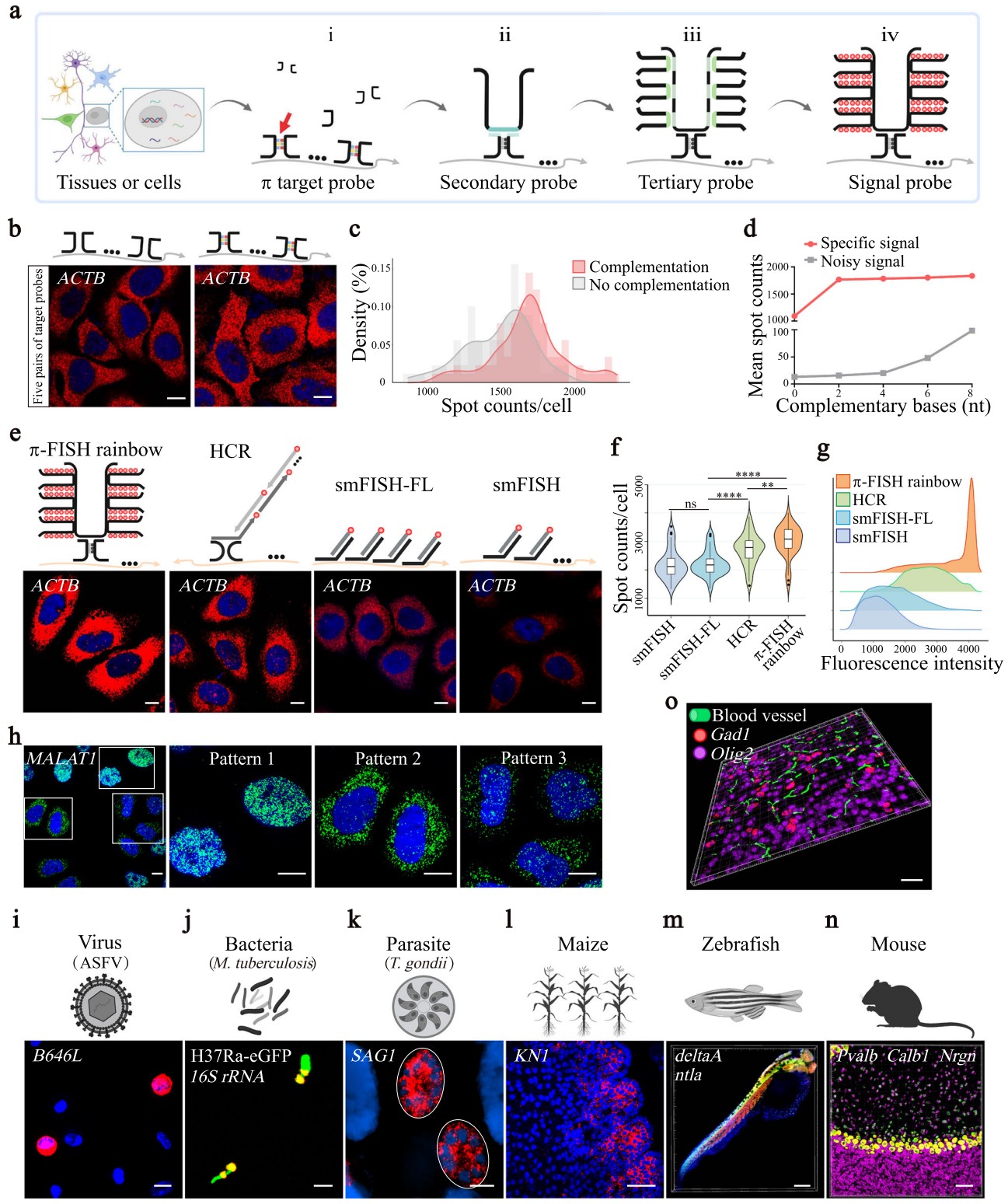

Comparing *ACTB* signal spot counts per cell detected by these methods showed that π-FISH rainbow had the highest sensitivity (Fig. 1f). The fluorescence signal intensity generated by π-FISH rainbow was also significantly higher than that generated by the other methods (Fig. 1g and Supplementary Fig. 6a–d). Similarly, we tested other medium-abundance (*PPIA* and *B2M* genes) and low-abundance (*MTOR* gene) transcripts using π-FISH rainbow, HCR, and smFISH, finally verifying that π-FISH rainbow was significantly more effective (Supplementary Fig. 6e).

To verify the specificity of π-FISH rainbow, we detected both high (*Cux2* and *Pcp4*) and low (*Lrmp* and *Ptpru*) expression genes in mouse brain tissues and reproduced the spatial distribution described by Allen Institute of Brain Science[27] (Supplementary Fig. 7a–d). The mutually exclusive gene expression patterns of *Sst* and *Vip* and of *Gad1* and *Slc17a7* revealed by π-FISH rainbow in the mouse cerebral cortex further supports the specificity of our method (Supplementary Fig. 7e–h). Next, we tested π-FISH rainbow specificity by detecting subcellular localization patterns of the long noncoding RNA (lncRNA)

**Fig. 1 | Highly sensitive and robust π-FISH rainbow for in situ detection of nucleic acid. a** Schematic of the π-FISH rainbow procedure. **b, c** Comparisons of hybridization efficiency between π target probes (with 2 complementary base pairs) and traditional split probes by detecting *ACTB* mRNA in HeLa cells (**b**). Scale bars, 10 μm. Spots/cell were counted and displayed as the histogram (**c**). *n* = 90 cells per group. **d** Effects of 0, 2, 4, 6, and 8 complementary base pairs on signal and noise were measured by detecting *PPIA* mRNA in HeLa cells. *n* = 50 cells per group. **e–g** Comparisons of hybridization efficiency among π-FISH rainbow, HCR, smFISH, and smFISH-FL by detecting *ACTB* mRNA in HeLa cells (**e**). Spot counts per cell (**f**) and fluorescent intensity per spot (**g**) were counted and illustrated. Scale bars, 10 μm. **f** π-FISH rainbow, *n* = 47 cells; HCR, *n* = 53 cells; smFISH-FL, *n* = 33 cells; smFISH, *n* = 40 cells. Data were expressed as mean ± s.e.m. Two-tailed unpaired Student's *t* test was used. **$P < 0.01$; ****$P < 0.0001$; ns, not significant. smFISH-FL vs. smFISH, $P = 0.8616$; HCR vs. smFISH-FL, $P = 1.26 \times 10^{-5}$; π-FISH rainbow vs. HCR,

$P = 0.0081$; π-FISH rainbow vs. smFISH-FL, $P = 1.04 \times 10^{-9}$. **g** π-FISH rainbow, $n = 8362$ spots; HCR, $n = 9564$ spots; smFISH-FL, $n = 14234$ spots; smFISH, $n = 14241$ spots. Box-plot with midline = median, box limits = Q1 (25th percentile)/ Q3 (75th percentile), whiskers = minimum and maximum values, points = outliers (>1.5 interquartile range). **h** Three subcellular localization patterns of *MALAT1* in HeLa cells were detected by π-FISH rainbow. Scale bars, 10 μm. **i–n** Detection of *B646L* mRNA of ASFV (**i**), 16S rRNA of *M. tuberculosis* (**j**), *SAG1* mRNA of *T. gondii* (**k**), *KN1* mRNA of maize ear (**l**). Codetection of *deltaA* (green) and *ntla* (red) in zebrafish embryo (**m**). Codetection of *Pvalb* (green), *Calb1* (red), and *Nrgn* (magenta) in mouse cerebellum (**n**) by π-FISH rainbow. Scale bars, 10 μm (**i**), 2.5 μm (**j**), 5 μm (**k**), 50 μm (**l**), 200 μm (**m**), and 50 μm (**n**). **o** Codetection of *olig2* and *Gad1* mRNA by π-FISH rainbow with vascular labeling in mouse brain. Scale bar, 50 μm. (**c, d, f, g**) Source data are provided as a Source Data file.

*MALAT1* in HeLa cells. Previous studies reported two localization patterns of this lncRNA: the majority of *MALAT1* was usually located in the nucleus, but the predominant signal was observed in the cytoplasm during the G2/M phase[28,29]. Our π-FISH analysis unambiguously demonstrated the nuclear pattern as well as the cytoplasmic pattern of lncRNA *MALAT1* in different cells (Fig. 1h). Owing to the high efficiency and specificity of π-FISH rainbow, we revealed, for the first time, a third localization pattern of lncRNA *MALAT1*: uniformly a low level of transcripts in both the cytoplasm and nucleus (9.1% of cells; Fig. 1h and Supplementary Fig. 7i, j).

To further demonstrate the robustness of π-FISH rainbow in different species, we first applied it for gene detection in microorganisms, including bacteria, viruses, and parasites. As shown in Fig. 1i, we successfully detected the African swine fever virus (ASFV) *B646L* gene expression in porcine alveolar macrophages. Due to the thick cell wall and lipid content of many bacteria, especially *Mycobacterium tuberculosis*, there are few effective FISH methods to detect gene expression in these bacteria. Here, we unambiguously identified the 16S rRNA of *M. tuberculosis* using π-FISH rainbow (Fig. 1j). We also detected the tachyzoite marker gene *SAG1* of *Toxoplasma gondii* in parasitophorous vacuoles (Fig. 1k). Furthermore, π-FISH rainbow was applied in plants for detection of *KN1* gene expression in paraffin-embedded maize ears (Fig. 1l). Next, we utilized π-FISH rainbow to detect *deltaA* and *ntla* mRNA in zebrafish whole-mount embryos (Fig. 1m). Finally, we detected the *Calb1*, *Nrgn*, and *Pvalb* mRNA to distinguish the molecular layer, Purkinje cell layer, and granule cell layer of the mouse cerebellar cortex in frozen sections (Fig. 1n). Our results demonstrated the versatility of π-FISH rainbow, which could be applied to diverse species in frozen, paraffin, and whole-mount samples.

Finally, we tested the compatibility of π-FISH rainbow with other imaging technologies, such as vascular labeling. To this end, biotinylated tomato lectin was employed to visualize blood vessels in the mouse brain. Subsequently, π-FISH rainbow was utilized for in situ gene detection. By this strategy, we could assess the spatial distribution of blood vessels, *olig2* (oligodendrocyte marker gene), and *Gad1* (inhibitory neuron marker gene) (Fig. 1o), suggesting that π-FISH rainbow is highly compatible with other imaging biotechnologies.

**Application of π-FISH rainbow in spatial cell type registration**

Single-cell sequencing has revolutionized the exploration of cell composition in tissues by identifying various marker genes for specific cell subtypes[3,5]. Here, we utilized π-FISH rainbow to further determine the spatial landscape of the cells in intact tissue by resolving the single-cell map of the marker genes (Fig. 2a). First, the spatial distribution of nine cortical layer-specific excitatory neuron marker genes in mouse primary somatosensory cortex (S1) was simultaneously determined using π-FISH rainbow through the combination of four fluorescence signal probes (Fig. 2b). We decoded the expression patterns of individual genes through channel arithmetic, cell segmentation, and spot calling using Imaris software (Fig. 2c and Supplementary Fig. 8).

As shown in Supplementary Fig. 9a, the layer-specific marker genes displayed a layer gradient distribution with a peak wave in the main expression layer. To further define the cortical sublayers, we rendered cells based on gene expression peaks, accurately reproducing the conventional layer-specific spatial pattern and precisely distinguishing the L5a and L6b sublayers (Fig. 2d).

Previous scRNA-seq analysis has classified *Gad1*[+] interneurons into multiple subclasses in the mouse S1[5]. We utilized 12 interneuron marker genes from the scRNA-seq data in S1 to test the capacity of π-FISH rainbow in the spatial registration of cell subclusters. First, we washed away the first-round hybridization signals of the nine layer-specific marker genes and performed a second-round hybridization for the 12 interneuron marker genes in the same tissue section (Fig. 2e). Based on the expression patterns of the interneuron marker genes, we resolved the spatial architecture of 13 *Gad1*[+] interneuron subclasses in S1 (Fig. 2f and Supplementary Fig. 9b). The excitatory and inhibitory neuron marker genes labeled distinct cells, as expected (Supplementary Fig. 9c). We further calculated the distribution of 13 *Gad1*[+] interneuron subclasses in the cortical sublayers based on the first-round layer-specific marker gene hybridization information. As shown in Fig. 2g, the Int 9 and Int 10 subclasses were mainly distributed in the superficial layers (L2/L3), whereas the Int 6 subclass was predominately located in the deep layers (L5 to L6). Together, these results demonstrated that π-FISH rainbow could be extensively applied in spatial cell type registration after scRNA-seq.

**Multiplexed in situ detection of proteins and neurotransmitters using π-FISH rainbow**

The throughput of conventional in situ detection of proteins by immunofluorescence is limited by the number of species of secondary antibodies. Moreover, for detecting low-expression proteins, enzyme amplification, such as the TSA system[30], is usually needed, but increasing background noise and reducing spatial resolution. To increase the throughput and sensitivity of in situ detection of proteins, we conjugated antibodies with different single-stranded DNA (ssDNA) and then employed π-FISH rainbow to detect the target ssDNA. This strategy comprises two steps: (i) Conjugation of ssDNA with specific barcodes to different antibodies and incubation of ssDNA-conjugated antibodies with the target proteins. (ii) The extension probe hybridization with the ssDNA and then signal amplification using π-FISH rainbow (Fig. 3a).

To evaluate the specificity, sensitivity, and background noise of this strategy, we expressed classical swine fever virus (CSFV) E2 protein in HeLa cells and compared conventional immunofluorescence (with secondary antibody) with the π-FISH rainbow strategy (Fig. 3b). Both π-FISH rainbow and the conventional immunofluorescence method showed similar staining patterns and signal variation trends in E2 protein-expressing cells (Supplementary Fig. 10a, b). Notably, analysis of the intensity surface plot demonstrated that π-FISH rainbow yielded much higher intensity fluorescence signals and less background noise (as indicated by the cells lacking the E2 protein in the oval region and

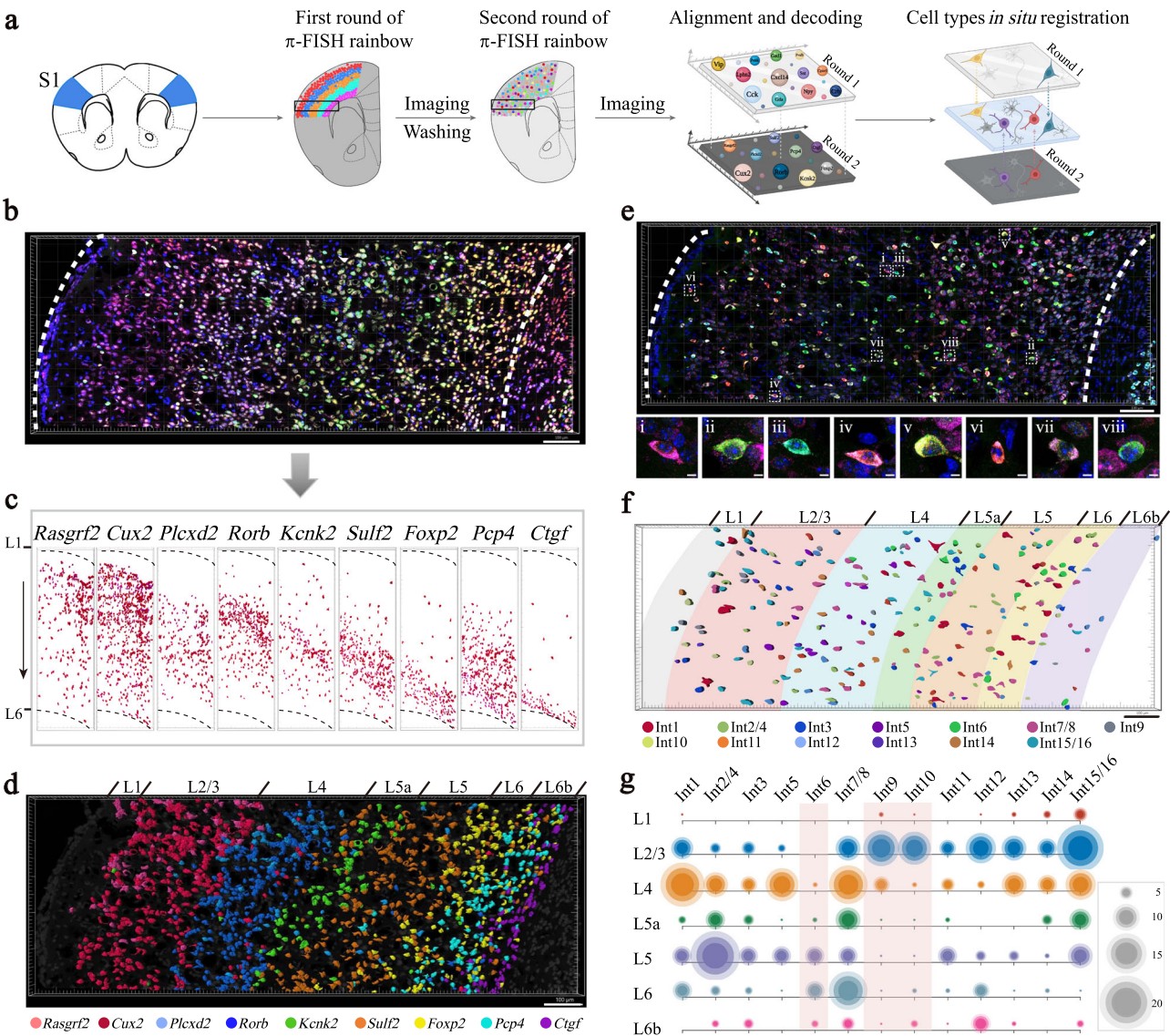

**Fig. 2 | Spatial registration of neuron subclasses in mouse primary somatosensory cortex (S1) by π-FISH rainbow multiplexed in situ detection. a** Diagram of spatial decoding of pyramidal neuron and interneuron subclasses in mouse S1 via two rounds of π-FISH rainbow hybridization and imaging. Nine excitatory neuron marker genes were detected in round 1, followed by signal wash-out and second-round hybridization of 12 interneuron marker genes. By combining two rounds of signals, spatial mapping of 13 interneuron subclasses in S1 were obtained. **b** Nine layer-specific excitatory neuron marker genes (*Rasgrf2, Cux2, Plcxd2, Rorb, Kcnk2, Sulf2, Foxp2, Pcp4,* and *Ctgf*) in S1 were simultaneously detected by π-FISH rainbow. White dashed lines indicated the boundaries of the cortex. Scale bar, 100 μm. **c** Decoding and mapping of signals from (**b**) showed the spatial distribution of nine neuron markers in the L1–L6 layers of the S1 cortex. The L1 and L6 of the cortex in S1 are indicated by the black dashed lines. **d** π-FISH rainbow accurately reproduced the spatial distribution of layer-specific marker genes, especially in the L5a and L6b

sublayers. Scale bar, 100 μm. **e** The second-round hybridization for 12 interneuron marker genes (*Gad1, Sst, Crh, Npy, Cck, Vip, Pvalb, Gda, Penk, Cxcl14, Cpne5,* and *Lphn2*) in the same section (top). (i-viii) The higher magnifications images were shown at the bottom. The boundaries of S1 are marked with white dashed lines. Scale bars, 100 μm (top) and 5 μm (bottom). **f** The distribution of 13 *Gad1*⁺ interneuron subclusters in different layers based on two-round hybridization. Scale bar, 100 μm. **g** Bubble plot for layer distribution of 13 *Gad1*⁺ interneuron subclasses in S1. Int 9 and Int 10 subclasses were mainly distributed in the superficial layers (L2/L3), whereas Int 6 subclass was predominantly located in the deep layers (L5 to L6) (light red frame). The number of interneuron subclasses is indicated by the spot size. n = 427 *Gad1*⁺ interneuron. Source data are provided as a Source Data file. Int1 to Int16 in (**f**) and (**g**) correspond to the interneuron subclasses defined by scRNA-seq from the publication by Zeisel et al.[5].

the regions without cells of the triangular region) (Fig. 3c and Supplementary Fig. 10a–c). We further tested the sensitivity of π-FISH rainbow in protein detection by primary antibody serial dilution. Our data showed that π-FISH rainbow could still detect the strong signal of RNA polymerase II (pol II) after 10,000-fold dilution, whereas the signal from traditional immunofluorescence was barely visible (Fig. 3d, e and Supplementary Fig. 10d–g). Importantly, noise signals in the negative controls were negligible (Supplementary Fig. 11a–h). Moreover, the false-positive rate of π-FISH rainbow was evaluated by co-detecting Pol II (endogenous protein) and CSFV E2 (exogenous

protein) in HeLa cells, and found to be 0.25% (Supplementary Fig. 11i). Together, these data validated the high sensitivity and specificity of the π-FISH strategy in protein detection.

To further test the capacity of π-FISH rainbow in multiplexed protein detection, we transfected IFNG (interferon-gamma)-flag into PK15 cells and infected the cells with CSFV and porcine circovirus 2 (PCV2). π-FISH rainbow simultaneously detected CSFV E2, PCV2 Cap, IFNG-flag, and pol II proteins and distinctly delineated the expression patterns of these four proteins in individual cells (Fig. 3f). As many small molecules have semi-antigen properties and can be specifically

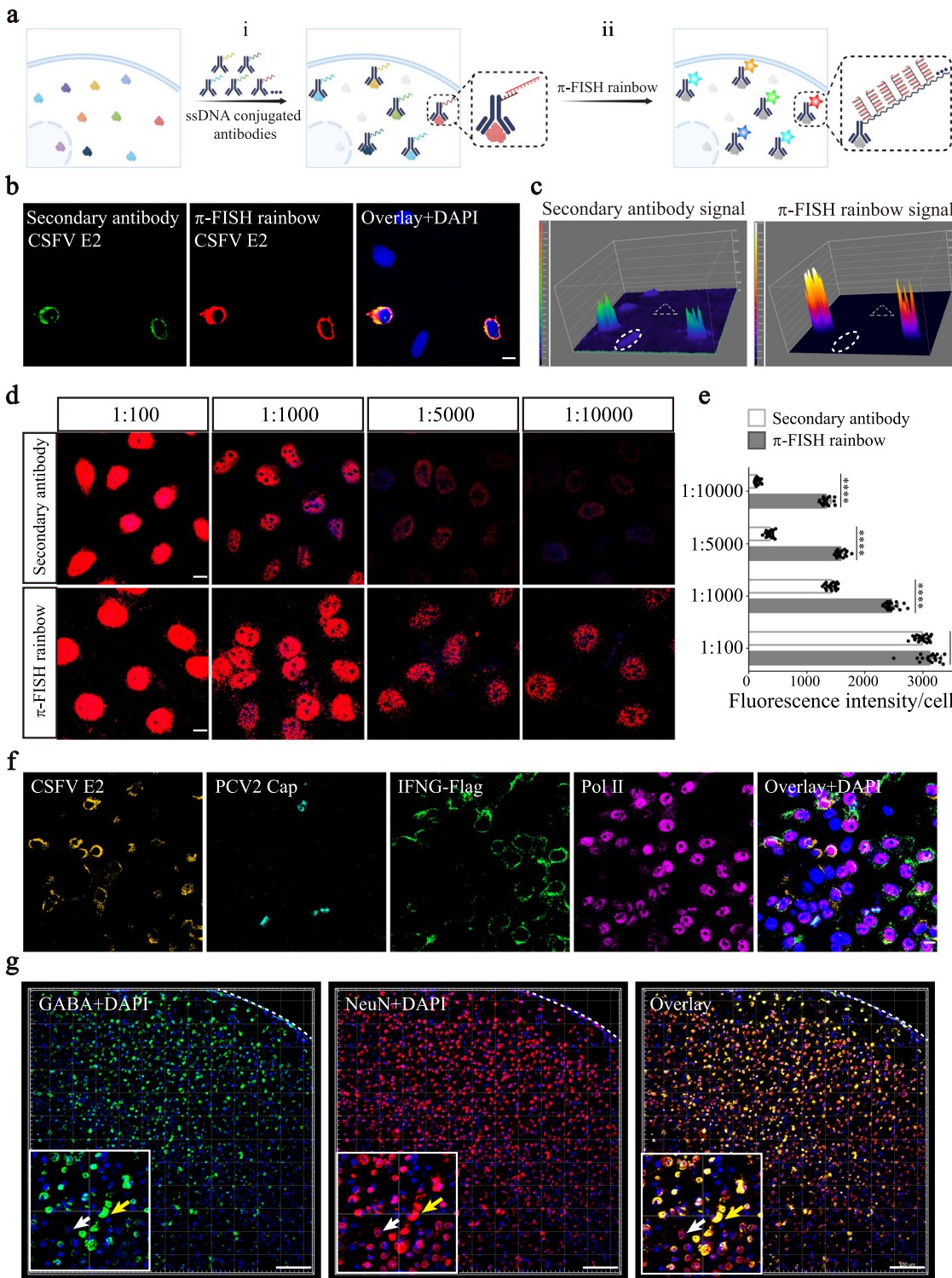

**Fig. 3 | In situ detection of proteins and neurotransmitters by π-FISH rainbow.**
**a** Schematic for in situ detection of multiple proteins and neurotransmitters by π-FISH rainbow. (i) Antibodies were conjugated to barcoded ssDNA and then incubated with target proteins and neurotransmitters. (ii) Hybridization of probes with barcoded ssDNA and signal amplification using π-FISH rainbow. **b** Detection of CSFV E2 protein in HeLa cells, transfected with the rAAV-E2 plasmid, by π-FISH rainbow and conventional immunostaining, respectively. Scale bar, 10 μm. **c** Intensity surface plot analysis of (**b**). Cells lacking E2 protein and background are indicated by oval region and triangular area, respectively. **d** Detection of pol II protein by conventional immunostaining (top) and π-FISH rainbow (bottom) with different dilutions of primary antibody (1:100, 1:1000, 1:5000, and 1:10,000),

respectively. Scale bars, 10 μm. **e** Quantification of mean signal intensity per cell of (**d**). *n* = 20 cells per group. Data were expressed as mean ± s.e.m. Two-tailed unpaired *t* test was used to compare two groups (1:100, *P* = 0.0137; 1:1000, *P* = 1.64E-29; 1:5000, *P* = 9.82E-38; 1: 10,000, *P* = 1.89E−39). *P < 0.05; ****P < 0.0001. Source data are provided as a Source Data file. **f** Simultaneous detection of CSFV E2, PCV2 Cap, IFNG-flag, and pol II proteins in CSFV and PCV2 co-infected PK-15 cells by π-FISH rainbow. Scale bars, 10 μm. **g** Codetection of GABA and NeuN in mouse cerebral cortex by π-FISH rainbow. Yellow arrow indicated cells co-expressing GABA and NeuN, and the white arrow indicated cells expressing only NeuN. The boundary of the cortex was indicated by white dashed lines. Scale bars, 150 μm.

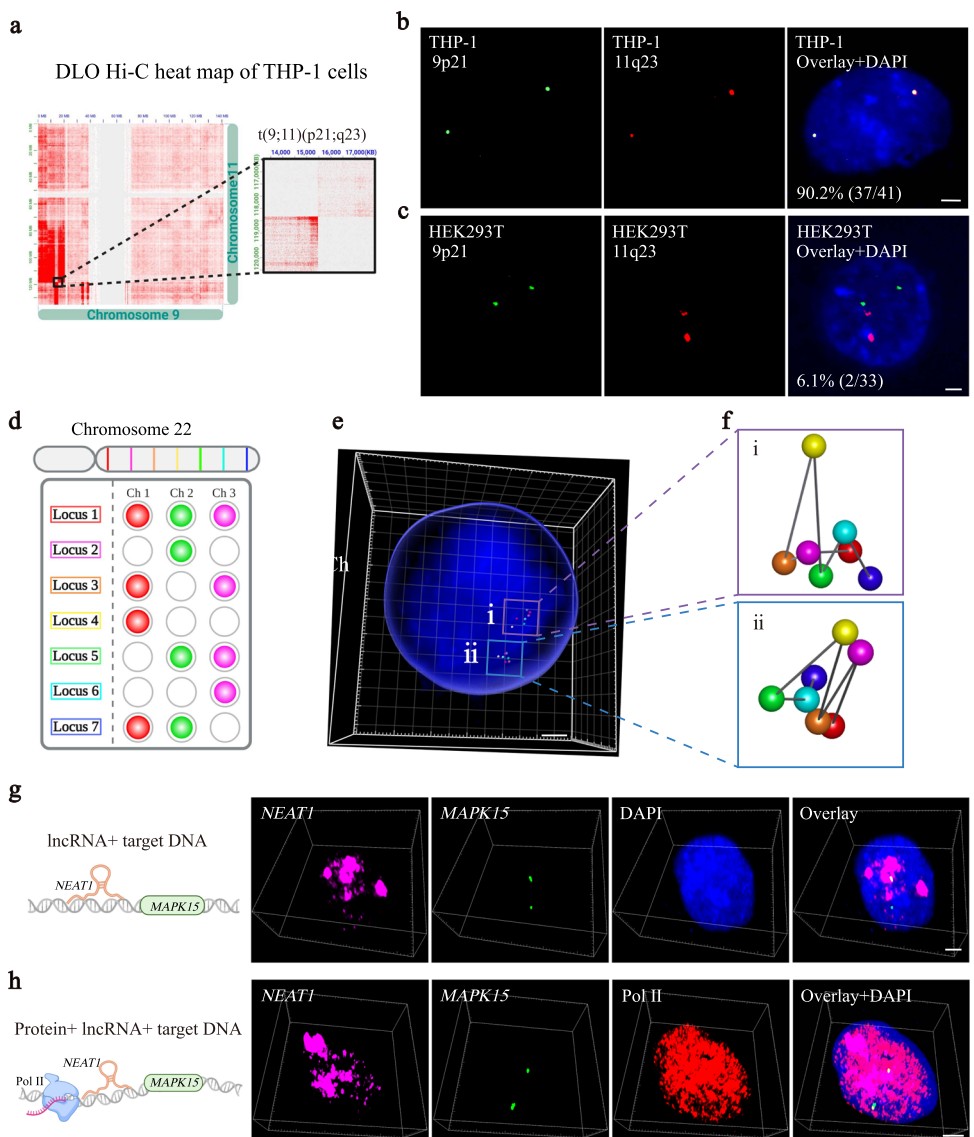

**Fig. 4 | Simultaneous in situ detection of DNA, RNA, and proteins by π-FISH rainbow. a** A putative translocation site between chromosomes 9 and 11 in THP-1 cells according to the DLO Hi-C data. **b, c** Detection of translocation sites in chromosomes 9 (green) and 11 (red) by π-FISH rainbow in THP-1 cells (**d**) and control HEK293T cells (**e**) confirmed the translocation in THP-1 cells. Scale bars, 2.5 μm. **d** Schematic for detecting seven loci on the long arm of chromosome 22 by π-FISH rainbow based on single and merged signals from three fluorescent channels: Ch 1, channel 1; Ch 2, channel 2; Ch 3, channel 3. **e** The hybridization signals for seven loci (rectangular box i and ii) on the long arm of chromosome 22 were decoded and reconstructed in the 3D image. Scale bar, 3 μm. **f** Homologous chromosome structure of the long arm of chromosome 22 based on hybridization signals. **g** Simultaneous detection of lncRNA *NEAT1* and its target *MAPK15* genomic locus in HeLa cells was imaged by confocal Z-axis scanning. *NEAT1* and *MAPK15* signal probes were labeled with Alexa Fluor 647 (magenta) and Alexa Fluor 488 (green), respectively. Scale bar, 2.5 μm. **h** Simultaneous detection of Pol II protein (red), *NEAT1* mRNA (magenta), and its target genomic locus (*MAPK15*, green) using π-FISH rainbow in HeLa cells followed by confocal Z-axis scanning. Scale bar, 2.5 μm.

recognized by antibodies, such as gamma-aminobutyric acid (GABA), an inhibitory neurotransmitter[31], we then employed π-FISH rainbow to simultaneously detect GABA and NeuN with ssDNA-conjugated antibodies. Figure 3g shows that the distribution of GABA in the mouse cerebral cortex was clearly detected by π-FISH rainbow, specifically in NeuN-positive cells. Overall, owing to its high sensitivity, specificity, and high-throughput capacity, π-FISH rainbow could be extensively used to decipher the spatial landscape of multiplexed proteins and small biomolecules in diverse tissues and organs.

## Simultaneous in situ detection of DNA, RNA, and protein by π-FISH rainbow

To test the application of π-FISH rainbow in DNA detection, we first designed one pair of π target probes to detect repetitive genome regions in telomeres and another to detect centromeres on chromosome 2 in HeLa cells. Both target probes yielded clear signals in the telomere and centromere regions (Supplementary Fig. 12a). The strategy could also identify polyploidy in HeLa cells (Supplementary Fig. 12b,c). We further targeted a nonrepetitive genomic locus (*Actb* genomic locus) in BHK cells and found that specific signals could be obtained using 35 π target probes (Supplementary Fig. 12d). Next, this strategy was employed to validate chromosomal translocation. To this end, we reanalyzed the DLO Hi-C data from THP-1 cells and identified a putative translocation between chromosomes 9 and 11 (Fig. 4a). To confirm the translocation, we designed π-FISH probes targeting the corresponding loci on chromosomes 9 and 11. Our data explicitly identified this translocation in THP-1 cells but not HEK293T cells serving as controls (Fig. 4b, c).

The 3D structure of the genome regulates many essential cellular functions, including DNA replication and transcription[32]. To directly visualize the 3D structure of the genome, we applied π-FISH rainbow for multiplexed genomic locus detection. First, two fluorescent signal probes for one genomic target were combined to test the feasibility of multiplexed signal coding. Our data showed that the two fluorescent signals targeting one locus overlapped (Supplementary Fig. 12e, f), suggesting the reliability of this method in multiplexed genomic locus decoding. We then simultaneously targeted seven loci on the long arm of chromosome 22 using combinatorial signal probes (Fig. 4d). The 3D spatial distributions of these seven loci are displayed in Fig. 4e. Next, we decoded hybridization signals and delineated the rough chromatin structure of the long arm of chromosome 22. Of note, we observed a substantial 3D structural difference between homologous chromosomes (Fig. 4f).

DNA, RNA, and protein usually form a dynamic complex to fulfill their physiological functions[33]. A simple and efficient method to simultaneously visualize the spatial-temporal organization of these complexes is desperately needed. Thus, we developed π-FISH rainbow to realize simultaneous codetection of different classes of biomolecules (DNA, RNA, and protein). It has been shown that mitogen-activated protein kinase 15 (MAPK15) may be the trans-binding site of the lncRNA nuclear enriched abundant transcript 1 (NEAT1)[34]. To directly confirm the interaction between MAPK15 and NEAT1, we designed π target probes specific for the MAPK15 genomic locus and NEAT1 RNA. Our π-FISH rainbow unambiguously demonstrated the spatially colocalized signals of MAPK15 and NEAT1 (Fig. 4g). To further test the capacity of π-FISH rainbow for multiplexed simultaneous detection of DNA, RNA, and proteins, we detected the spatial distribution of the lncRNA NEAT1, Pol II protein and the genomic locus of MAPK15 in HeLa cells (Fig. 4h). These high-quality FISH signals for different biomolecules suggest the broad potential application of π-FISH rainbow in the multiomics investigation.

## Development of π-FISH+ to detect microRNA, alternative splice variants, and short DNA loci

Current FISH methods require a length of approximately 500 bp[23] for target regions to provide enough space for multiplex probes to yield a strong hybridization signal. It is still challenging to detect short-length targets, such as microRNAs, alternative splice variants, and short DNA loci, with sufficient signal intensity and specificity by a single pair of probes. Thus, we developed π-FISH+, which combines π-FISH rainbow with HCR technology to achieve intensive amplification using only one pair of π target probes. As illustrated in Fig. 5a and Supplementary Fig. 13, the signal probe in step 4 of π-FISH rainbow is replaced with the HCR split-initiator probes and self-folding hairpins. Each tertiary amplification probe hybridizes with four pairs of split-initiator probes. Thus, π-FISH+, using one π target probe can theoretically initiate 64 HCR reactions to generate sufficient signal amplification for detection and imaging.

To verify the variation in signal intensities and fluorescence spot sizes, we performed HCR amplification using one pair of π target probes for PPIA for 1, 2, 4, 8, and 16 h to generate varying amplification polymers (Supplementary Fig. 14a). HCR amplification for 16 h yielded maximal signal brightness without increasing spot size, meaning that it could be used for π-FISH+ detection (Supplementary Fig. 14b, c). Notably, we found that the coefficient of variation was similar for all reaction times, indicating that the HCR amplification degree does not affect the spot brightness (Supplementary Fig. 14d). Notably, the π-FISH+ signal for PPIA is highly specific since almost no noise signal was detected in the negative controls (Supplementary Fig. 14e). To calculate the false-positive rate, we co-detected PPIA and bacterial dapB (not expressed in eukaryote) in HeLa cells using π-FISH+, and found it to be 0.36% (Supplementary Fig. 14f, g). The detection efficiency of π-FISH+ was similar to that of π-FISH (Supplementary Fig. 14h, i). However, the

fluorescence intensity of the spot was significantly higher than that of π-FISH rainbow (Supplementary Fig. 14j).

We first applied π-FISH+ to detect miR145-5p and its sponge lncRNA MALAT1[35]. As miR145-5p contains only a 23-bp seed sequence, a 54-bp ssDNA extension probe was hybridized to the seed sequence to generate sufficient space for the π-FISH+ amplification system. The 54 bp extension probe was shown to be necessary and specific for miR145-5p detection since noise signals were negligible for no probe or scramble extension probe controls (Supplementary Fig. 15a–c). Additionally, the expression pattern of MALAT1 verified by π-FISH+ was consistent with the π-FISH rainbow results, further proving the efficiency and accuracy of the π-FISH+ strategy (Supplementary Fig. 16a, b). π-FISH+ codetection showed that aggregated MALAT1 in the nucleus was colocalized with miR145-5p (Fig. 5b and Supplementary Fig. 16c).

Previous studies indicated that the expression of the androgen receptor splicing variant ARV7 in CTCs of prostate cancer patients is associated with innate and acquired resistance to the androgen receptor (AR)-targeted therapies[36]. Although numerous attempts have been made to distinguish ARV7 at the protein or mRNA levels from other splice variants, the efficiency and accuracy of these approaches remain ambiguous. Thus, we applied π-FISH+ to identify ARV7 by simultaneously labeling ARV7 locus 1 (transcript in ARV1, ARV2, ARV4, and ARV7) and locus 2 (transcript in ARV5 and ARV7) using two fluorescence signal probes. In this scenario, the specific signal corresponding to ARV7 was obtained when two fluorescence signals were colocalized (Fig. 5c). To test this, we examined ARV7 and K19 (a marker of CTCs) in two well-established prostate cancer cell lines, LNCaP (with ARV7 expression) and PC3 (without ARV7 expression). As shown in Fig. 5d, e, ARV7 was unambiguously observed in LNCaP cells but was almost absent in PC3 cells. To validate the capability of π-FISH+ in clinical diagnosis, we further applied π-FISH+ in CTCs from the blood of prostate cancer patients and successfully detected ARV7 variants (Fig. 5f, g and Supplementary Fig. 17), indicating the integrity of this strategy for anti-androgen therapy resistance diagnosis.

Furthermore, we applied π-FISH+ to detect small genomic indels and breakpoints. To this end, we deleted the DNA sequence on chromosome 12: 40360484–40361854 in A549 cells by the CRISPR/Cas9 system and generated an A549-KO cell line. Next, one pair of π target probes was designed to target locus 1 as a reference, and another was designed to target the breakpoint, locus 2, as illustrated in Fig. 5h. Our data demonstrated that the signals corresponding to locus 1 and locus 2 overlapped in A549 wild-type (A549-WT) cells. In A549-KO cells, only the reference green fluorescence signal spots were detected (Fig. 5i). These results suggest that π-FISH+ can be applied to detect small DNA indels and breakpoints by only one pair of π target probes and may be potentially applied for quick and straightforward hereditary disease diagnosis.

## Discussion

FISH is a classical and indispensable method for molecular biology research and precise diagnosis[37]. Despite the recent upgrade of this technology, the new generation of cutting-edge FISH technology is still in its infancy. Here, we developed π-FISH rainbow, a high-efficiency multiplex FISH method with high-intensity signals and low background noise. This approach was successfully applied to detect diverse biomacromolecules (DNA/RNA/protein) and neurotransmitters individually or simultaneously. Furthermore, we combined π-FISH rainbow with HCR[14] to develop π-FISH+ technology to detect short nucleic acid fragments, such as microRNA and transcript variants. These methods can be widely applied in animals, plants, and pathogenic microorganisms, providing powerful means for clinical diagnosis and biological research.

To increase hybridization stability and efficiency, we upgraded the conventional split probe with 2–4 complemented base pairs in

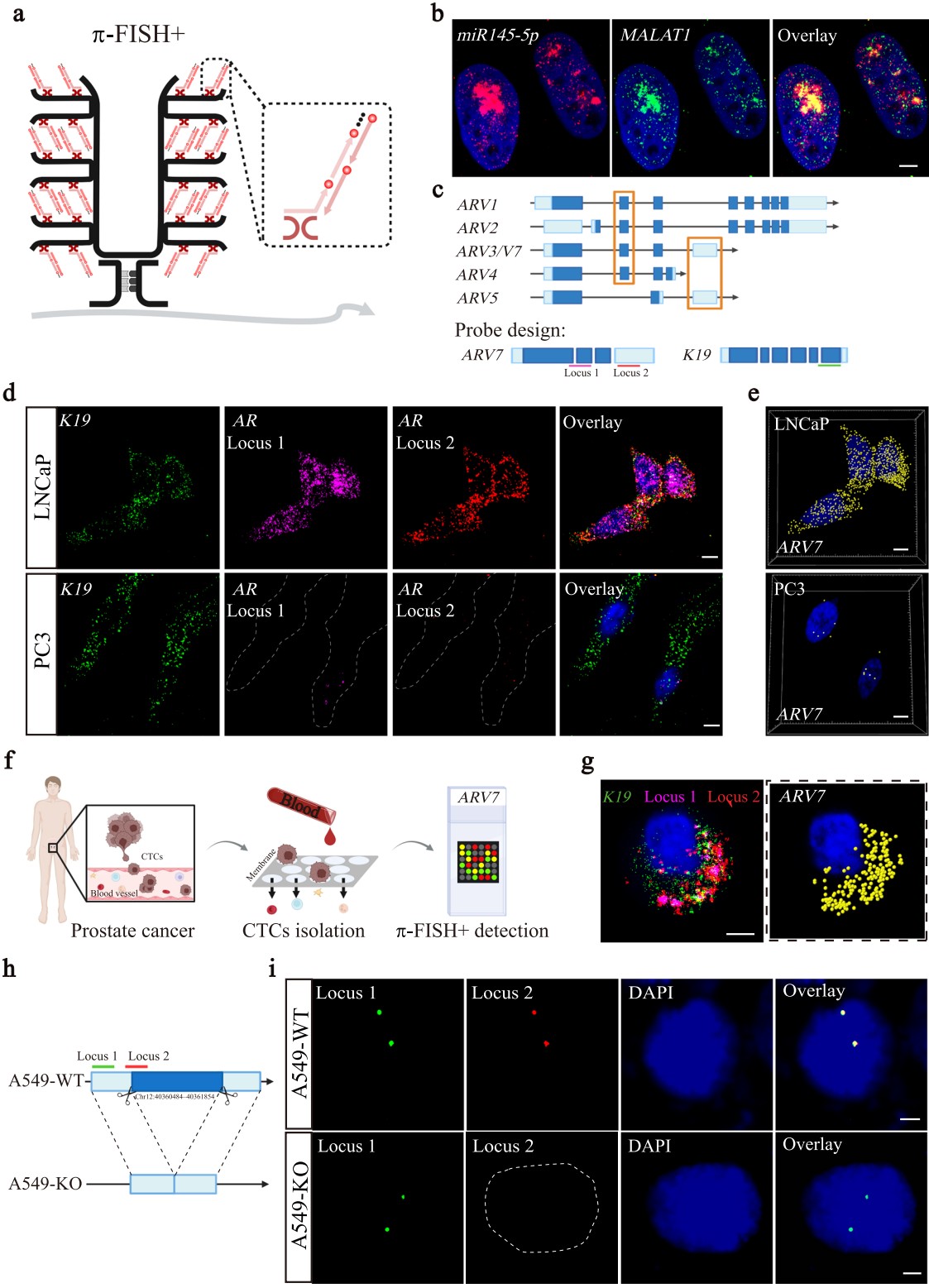

the middle bond region to form a more stable structure. Our data demonstrated that π-FISH rainbow could indeed facilitated hybridization and significantly improved the efficiency and specificity of the signal from highly expressed housekeeping genes (e.g., *ACTB*) to low-profile genes (e.g., *MTOR*). Compared with those of HCR[14] and smFISH, the π-FISH rainbow signal intensity and fluorescent spot counts were significantly higher. Although RCA can also generate high-intensity signals, efficient enzymatic reaction efficiency in complex biological tissues remains a challenge[19,26]. Previous FISH

methods identified only two subcellular localization patterns for lncRNA *MALAT1*, either in the nucleus or the cytoplasm[28,29,38]. Owing to the high sensitivity of π-FISH rainbow, we discovered a new subcellular distribution pattern of *MALAT1* with uniformly low expression in both the nucleus and cytoplasm, demonstrating the power of π-FISH rainbow in gene transcription research. While 1–5 probe pairs per gene are enough to detect highly abundant transcripts, we recommend using 10–15 probe pairs to detect medium or low-expressed transcripts.

**Fig. 5 | Development of π-FISH+ to detect short nucleotide sequences, including microRNAs, alternative splice variants, and short DNA variations.**
**a** Schematic of π-FISH+: The signal probe of π-FISH rainbow was replaced with the HCR split-probe for the hybridization chain reaction to amplify the signal further. **b** The *miR145-5p* and its sponge lncRNA *MALAT1* were co-detected by only one pair of π target probes for each gene by π-FISH+. Scale bar, 5μm. **c** Diagram of probe design for identifying the *ARV7* splice variant: The variant could be identified by simultaneously labeling locus 1 (in *ARV1*, *ARV2*, *ARV4*, and *ARV7*) and locus 2 (in *ARV5* and *ARV7*). The signal of *K19* gene was used as an internal control.
**d** Codetection of locus 1 (magenta), locus 2 (red), and *K19* (green) in prostate cancer cell lines LNCaP (with *ARV7* expression) and PC3 (without *ARV7* expression). *K19* was expressed in both cell lines as a control. Dashed lines indicated cell boundaries. Scale bars, 5 μm. **e** 3D rendered image of *ARV7* signal in LNCaP (top) and PC3

(bottom) cells from (d). Positive signals for *ARV7* are indicated with yellow spots. Scale bars, 5 μm. **f** Application of π-FISH+ for *ARV7* detection in CTCs for clinical diagnosis during prostate cancer therapy. **g** The *ARV7* variant was successfully detected in CTCs from prostate cancer patient blood. Scale bar, 2.5 μm. **h** Diagram of probe design for detecting small genomic indels and breakpoints. One pair of π target probes was designed to target locus 1 as a reference, and another was designed to target the breakpoint, locus 2. A549-KO cell line was generated by deleting the DNA sequence in chromosome 12: 40360484–40361854 in A549 cells using the CRISPR/Cas9 system. **i** Signals corresponding to locus 1 (green) and locus 2 (red) overlapped in A549 wild-type (A549-WT) cells, while only locus 1 was detected in A549-KO cells. Dashed lines indicated cell boundaries. Scale bars, 2.5 μm.

Despite recent progress in FISH technologies, they require long nucleic acid fragments for multiplexed target probes in each site to generate sufficiently strong signals. It remains a great challenge to detect microRNAs, alternative splice variants, and DNA indels. In this regard, we developed a π-FISH+ method by combining π-FISH rainbow with HCR techniques to achieve sufficient amplification capacity and simultaneously guarantee signal specificity. Our data demonstrated that π-FISH+ could effectively detect the distribution of *miR145-5p* and its colocalization with its target lncRNA *MALAT1*[35]. We also used π-FISH + to verify the loss of short DNA indels in A549-KO cells generated by CRISPR editing, supporting its robustness in small fragment RNA/DNA detection. Of note, we successfully distinguished *ARV7* from other splicing variants, which is greatly desired for precise diagnosis during clinical prostate cancer therapy. In principle, π-FISH rainbow can also be combined with other approaches to amplify signal intensity, such as SABER[16], making it a flexible tool to investigate spatial gene expression on a variety of platforms.

Biomacromolecules (e.g., protein, RNA, and DNA) or neurotransmitters often interact to form a complex to fulfill their physiological roles in a precise spatial-temporal manner[13,31,39,40]. Delineation of the specific spatial distribution and the content of diverse biomacromolecule complexes is pivotal to understand the regulation of various biological processes[41,42]. Although many FISH methods can co-detect DNA, RNA, and proteins[13,24], these methods rely on multiple rounds of hybridization, imaging, and alignment. Thus, an efficient and accurate simultaneous detection method of genomic loci, RNA transcripts, and proteins in one-step imaging is desperately needed. To this end, we adapted the π-FISH rainbow protocol for protein, RNA, and DNA codetection and demonstrated the colocalization of Pol II, *NEAT1*, and its regulatory genomic target *MAPK15* gene[34]. Such a codetection method can provide critical insights into the relationships between biomolecular interactions and elucidate their regulatory mechanisms. Furthermore, π-FISH rainbow is versatile and highly compatible with other imaging technologies, such as vascular labeling using biotinylated tomato lectin, which revealed the comprehensive spatial distribution of vasculature, oligodendrocytes, and neurons. With the robustness of π-FISH rainbow, it is expected that this method will be extensively combined with additional imaging technologies, such as $Ca^{2+}$ imaging[43] and Raman imaging[44], to decipher multidimensional spatial omics.

The rapid advent of single-cell sequencing has yielded unprecedented information about new subclusters of cell types with diverse marker genes[5,45–47]. A highly sensitive and cost-effective FISH method with medium throughput could provide ample opportunities to delineate the spatial landscape of cell types in various organs. Here, we employed π-FISH rainbow to map the spatial distribution of diverse neuron subtypes in different sublayers by decoding the spatial distribution of 21 marker genes from scRNA-seq data[5] by π-FISH rainbow. Our data demonstrated that the integrated analysis of π-FISH rainbow and scRNA-seq data could be extensively used for in situ annotation of cell types in tissues. Here, we deciphered only 21 genes in two rounds

as proof of concept. It should be possible to detect more genes with more rounds of hybridization. To further increase the throughput, π-FISH rainbow could be amplified by probes with distinct DNA barcodes and decoded by in situ sequencing or multiplex sequential hybridization in prospective studies. In this way, π-FISH rainbow will enable exponentially higher throughput while maintaining its sensitivity and robustness.

Overall, we developed a high-efficiency, versatile and robust multiplex FISH method, π-FISH rainbow, with high-intensity signals and low background noise. We demonstrated the application of this robust FISH method in animals, plants, and pathogenic microorganisms. Of note, π-FISH rainbow can be utilized to visualize the chromosome spatial conformation and co-detect diverse biomacromolecules (DNA/RNA/protein) and neurotransmitters, which may greatly facilitate deciphering the molecular machinery and regulatory mechanism of molecular complexes in cells. By combining π-FISH rainbow with HCR, we realized the detection of short nucleic acid fragments, such as microRNA and prostate cancer anti-androgen therapy-resistant marker *ARV7* splicing variant. Our study provided a powerful tool for in situ detection of diverse biomolecules and could be extensively used in various biological investigations and clinical diagnoses.

## Methods
### Ethical statement
This study was conducted according to the guidelines for experimental animals of the Research Ethics Committee of Huazhong Agricultural University. The use of animal and human blood samples was approved by the Scientific Ethic Committee of Huazhong Agricultural University (HZAUMO-2022-0063; 202204060001).

### Cell, bacteria, parasite, experimental animals, and human blood samples
HeLa (ATCC CCL-2), HEK293T (ATCC, CRL-1573), BHK (ATCC, PTA-4506), A549 (ATCC, CCL-185), human foreskin fibroblast (HFF) (ATCC, SCRC-1041), LNCaP (ATCC, CRL-1740), PK-15 (ATCC, CCL-33), and PC3 (ATCC, CRL-1345) cell lines were cultured in Dulbecco's Modified Eagle Medium (DMEM, Gibco, 11995073) supplemented with 10% fetal bovine serum (FBS; Gibco). THP-1 (ATCC, TIB-202) cells were cultured in RPMI 1640 (Gibco, 11875119) supplemented with 10% FBS. Porcine alveolar macrophages (PAM) were prepared by bronchoalveolar lavage as described[48]. All cells were cultured at 37 °C in the presence of 5% $CO_2$. *M. tuberculosis* H37Ra-eGFP used in this study was obtained from the American Type Culture Collection (ATCC25177) and was cultured in Middlebrook 7H9 broth (Becton, Dickinson) supplemented with 10% oleic acid albumin dextrose catalase, 0.05% Tween-80, and 0.5% glycerol. RH strain of *T. gondii* (a gift from Bang Shen, Huazhong Agricultural University) was maintained in HFF cells by serial passage. Wild-type CD-1 (male, 4 weeks old) and Wild-type C57/BL6 (male, 8 weeks old) mice were housed at 22–25 °C, 40–60% humidity, and a dark/light cycle of 12/12 h. Wild-type AB

zebrafish (*Danio rerio*) were grown in water at 28 °C under a 14:10-h light-dark photoperiod. Fertilized eggs were collected within 30 min after fertilization and incubated in Petri dishes until 27 h post-fertilization (hpf). 5 mL of blood from each of three prostate cancer patients aged 58–65 were collected for CTCs isolation. We had informed consent from the patients.

## Injection of tomato lectin and tissue fixation

To label vascular elements, a total of 100 μg/kg of tomato lectin Dylight 488 (Thermo Fisher Scientific, L32470) from *Lycopersicon esculentum* was intravascularly injected into anesthetized C57/BL6 mice. Brains were extracted 5 min after the injection and placed in cold 4% (w/v) paraformaldehyde for 24 h, dehydrated for 12 h using 30% sucrose, embedded in OCT, and processed into frozen sections following standard procedures for hybridization.

## Cells, ASFV, *M. tuberculosis*, and *T. gondii* sample preparation

Cover slips were washed with PBS for 1 min and treated with 100 μg/mL poly-D-lysine (Thermo Fisher Scientific, A3890401) for 10 min. The poly-D-lysine was discarded and the cover slips were dried in a 48-well culture plate. The cells were washed with PBS and treated with 0.25% (w/v) trypsin (Gibco, 25300120). After being resuspended in the culture medium, the cells were seeded on the cover slips (poly-D-lysine treated) and cultured for 24 h for in situ detection. *T. gondii* was cultured in HFF cells grown to 90% confluence on the cover slips for 20 h to form the parasitophorous vacuole (PV). To confirm the ability of π-FISH rainbow to detect viruses, PAM cells were infected with ASFV at the multiplicity of infection (MOI) of 0.1 for 24 h before hybridization. Samples were washed with PBS for 5 min and then fixed with 4% (w/v) paraformaldehyde for 10 min. Samples were eluted with gradient ethanol (70%, 85%, and 100%) for 3 min each, permeabilized with 0.2 M HCl for 5 min, and then subjected to 5 μg/mL proteinase K (Sigma-Aldrich, 3115836001) for 2 min. *M. tuberculosis* required additional treatment with 5 μg/mL lysozyme (Thermo Fisher Scientific, 90082) at 37 °C for 2 h following HCl treatment. Samples were fixed again for 5 min using 4% (w/v) paraformaldehyde and washed three times with PBST for 3 min. RNase-free reagents were used for RNA detection. Then, the samples were prepared for the following hybridization for π-FISH rainbow.

## Cryo-embedded tissue section preparation

Mice were anesthetized and transcardially perfused with 4% (w/v) paraformaldehyde. Brain samples were immediately fixed with cold 4% (w/v) paraformaldehyde for 24 h, dehydrated for 12 h using 30% sucrose, embedded in optimal cutting temperature compound (OCT, SAKURA, 4583), and processed into frozen sections following standard procedures. A series of frozen brain sections (10–12-μm thick) were fixed with 4% (w/v) paraformaldehyde for 15 min, washed three times with PBST for 5 min, and then dehydrated in an ethanol series. The tissue was permeabilized in 0.2 M HCl for 10 min and 5 μg/mL proteinase K for 5 min. Finally, the tissue was fixed with 4% (w/v) paraformaldehyde for 10 min and washed three times with PBST for 5 min each for the next hybridization.

## Paraffin-embedded maize ear preparation

Paraffin-embedded maize ear samples were cut into 8-μm thick sections for xylene dewaxing and gradient ethanol dehydration following standard procedures. They were then treated with 1% (w/v) cellulase (Yakult, Onozuka™ R-10) and 0.5% (w/v) pectinase (Yakult, Macerozyme™ R-10) for 10 min and washed three times with PBS for 3 min each. Sections were permeabilized with 0.2 M HCl for 10 min and 5 μg/mL proteinase K for 2 min. Finally, the tissue sections were fixed with 4% (w/v) paraformaldehyde for 10 min and washed three times with PBST for the subsequent hybridization.

## Zebrafish whole-mount embryo preparation

Zebrafish whole-mount embryos of 27 hpf were fixed with 4% (w/v) paraformaldehyde at 4 °C for 24 h. Samples were washed with PBS three times for 3 min and stepwise transferred to a series of methanol (25%, 50%, 75%, and 100%) for 5 min each. Embryos were placed in fresh methanol and stored at −20 °C until use. When preparing for in situ hybridization, the embryos were treated with gradient methanol (75%, 50%, and 25%) for 5 min each at room temperature. Then, embryos were permeabilized with 0.2 M HCl for 10 min and subjected to proteinase K (Beyotime Biotechnology, ST533; 5 μg/mL in PBS) for 5 min at room temperature. After being fixed again with 4% (w/v) paraformaldehyde for 5 min, the embryos were washed three times with PBST. Then, the samples were prepared for the following hybridization.

## Probe design of π-FISH rainbow

The π-FISH rainbow amplification system consists of a four-step hybridization process.

Step 1: the π target probes contain 2–4 complementary base pairs in the middle region, which form a π shape. Half of the π target probe consists of three sections: bottom target region (20–25 nucleotides (nt) for hybridization with target genes), top region (14 nt for hybridization with secondary probes), and middle region (8 nt, 2–4 nt of which are designed to form bonds between π target probes). Step 2: the secondary U-shaped amplification probes (509 nt) consist of two sections: the middle region and the 5′ and 3′ arm regions. The middle region (29 nt) hybridizes with the top region from the left and right sides of the π target probe pairs. If half of the π target probe binds with a non-specific sequence, the secondary probe is easily washed away under a series of strict washing conditions to avoid signal noise. The 5′ and 3′ arm regions consist of 16 repeating sequences, with each repeat containing a 20 nt hybridization region for binding of the tertiary probes and a 10 nt spacer region. Step 3: The tertiary U-shaped amplification probes (260 nt) consist of two sections: the middle region and the 5′ and 3′ arm regions. The middle region (20 nt) is designed to hybridize with the repeated regions of the 5′ and 3′ arms of the secondary amplification probes. The 5′ and 3′ arm regions consist of 8 repeating sequences, each containing a 20 nt hybridization region for binding signal probes and a 10 nt spacer region. Step 4: The 20 nt signal probe is conjugated with fluorophores at both 5′ and 3′ ends and is them used to hybridize with repeat regions of the tertiary amplification probes. The signal probes are conjugated with Alexa Fluor 488, Alexa Fluor 546, Alexa Fluor 594, or Alexa Fluor 647.

When detecting the co-expression of multiple genes, different combinations of signal probes are used to improve the throughput of gene detection. Theoretically, combining four fluorescent signal probes can generate 15 ($C^1_4 + C^2_4 + C^3_4 + C^4_4$) signal codes and decode 15 genes in one go. To generate multiplex signals for one RNA, π target probes were divided into different groups. Each group's individual π target probes were hybridized with the same secondary probe. Combining different groups of probes allows digital encoding of the multiplex signal. The number of π target probes per mRNA increases based on the number of fluorescent colors, generating a consistent luminance corresponding to each fluorescence signal. All probe sequences are listed in Supplementary Data 1.

## smFISH and hybridization chain reaction (HCR)

The probe preparation and experiment procedure of smFISH[9] and HCR[14] were performed as previously described. For smFISH, prepared samples were incubated in hybridization solution (2× SSC (Ambion, AM9763), 30% (v/v) formamide (Ambion, AM9342), 2 mM VRC (NEB, S1402S), 1 mg/mL yeast tRNA (Life Technologies, 15401-011), 10% (w/v) dextran sulfate (Sigma-Aldrich, D8906), and 10 nM target probes) for 18–36 h at 37 °C. Then, samples were washed three times with washing

buffer (2× SSC, 30% (v/v) formamide, 2 mM VRC). Next, samples were incubated in hybridization buffer (2×SSC, 10% (v/v) formamide; 10% (w/v) dextran sulfate, and 2 mM VRC) with 100 nM fluorescently labeled probes for 15 min at 37 °C. Finally, samples were washed three times with washing buffer and imaged immediately. For HCR, prepared samples were incubated in hybridization solution (5× SSC, 30% (v/v) formamide, 9 mM citric acid (pH 6.0, Thermo Fisher Scientific, 005000), 0.1% Tween 20 (Sigma-Aldrich, P1379), 50 µg/mL heparin (Sigma-Aldrich, H3149), 10% (w/v) dextran sulfate, and 10 nM target probes) overnight (12–16 h) at 37 °C. Samples were then washed twice with washing buffer (5× SSC, 30% (v/v) formamide, 9 mM citric acid (pH 6.0, Thermo Fisher Scientific, 005000), 0.1% Tween 20, 50 µg/mL heparin) for 5 min at room temperature. Next, samples were incubated in amplification solution (5× SSC, 0.1% Tween 20, 10% dextran sulfate, and 100 nM amplification probes) overnight (12–16 h) at room temperature. Last, samples were washed three times with 5×SSCT and imaged immediately. The probe sequences of smFISH and HCR can be found in Supplementary Data 1.

## Probe synthesis and preparation

The π target probe and signal probes are synthesized DNA oligonucleotides. Secondary and tertiary amplification probes are ssDNA probes generated from plasmid templates through in vitro transcription and reverse transcription. All π target probes were under 60 nt and synthesized by GenScript company. To construct plasmids for secondary and tertiary amplification probes, a T7 promoter sequence (TAATACGACTCACTATAGGG) and terminator sequence (CTAGCATA ACCCCTTGGGGCCTCTAAACGGGTCTTGAGGGGTTTTTT) were added upstream and downstream of the secondary and tertiary probe sequences, respectively. A 30-nt reverse primer sequence (TGCTA GCCCATGATCGTCCGATCTGGTCGG) was added to the end of each sequence to facilitate probe construction. The designed sequence was inserted into the pUC19 vector via gene synthesis to obtain final secondary and tertiary amplification probe plasmids. One microgram of the secondary and tertiary probe plasmid template was used in the in vitro transcription reaction using HiScribe™ T7 High Yield RNA Synthesis Kit (NEB, E2040S). To generate cDNA from the RNA product, reverse transcription reactions were carried out using Thermo Scientific™ Maxima™ H Minus Reverse Transcriptase (Thermo Fisher Scientific, EP0752) for 60 min at 50 °C, and the reaction was terminated by heating at 85 °C for 5 min. Finally, 10 µL of 0.5 M EDTA and 10 µL of 1 M NaOH were mixed, added to 20 µL cDNA product, and then incubated at 95 °C for 10 min to remove the RNA template. The cDNA product was purified using Oligo Clean & Concentrator™ kit (Zymo Research, D4060). The probes were eluted into 50 µL ultrapure water, quantified, and stored at −20 °C. For the signal probe, both 3′ and 5′ ends were conjugated with Alexa Fluor series fluorescence.

## Multiplexed RNA in situ detection by π-FISH rainbow

The multiplexed RNA in situ detection experiment was conducted following a four-step procedure. First, the cells or tissues were incubated in hybridization solution A (10% (w/v) dextran sulfate (Sigma-Aldrich, D8906), 6× saline-sodium citrate (SSC; Ambion, AM9763), 25% (v/v) formamide (Ambion, AM9342), 0.3% lithium dodecyl sulfate (LDS; Sigma-Aldrich, L9781), 10% (w/v) Denhardt's solution (Thermo Fisher Scientific, 750018), and 1% vanadyl-ribonucleoside complex (VRC; NEB, S1402S) for 8–10 h at 40 °C. Next, samples were incubated in hybridization solution A containing the π target probe pair at final concentrations of 10 nM. Samples were then washed gently by a series of gradient elution buffers to minimize noise and ensure positive signals, including elution buffer A (consisting of 2× SSC, 20% (v/v) for-mamide, and 0.03% LDS) preheated to 40 °C for 5 min, then washed twice with elution buffer B (1× SSC, 10% (v/v) formamide and 0.03% LDS) for 5 min and finally washed with elution buffer C (0.1× SSC, 0.03% LDS) for 5 min.

In the second step, hybridization solution B (10% (w/v) dextran sulfate, 5× SSC, 20% (v/v) formamide, 0.3% LDS, 10% (w/v) Denhardt's solution, and 1% VRC) was preheated to 40 °C. The secondary probe was denatured by heating at 85 °C for 2 min and then cooling on ice for 5 min. Next, the samples were incubated with hybridization solution B and 10–50 nM of the secondary probe for 2–3 h. Samples were then eluted with elution buffer A, B, and C as in Step 1.

In the third step, the samples were incubated with hybridization solution B and 10–50 nM of the denatured tertiary probe (preheated to 40 °C) for 2–3 h at 40 °C. The tertiary probe was denatured before adding to solution B. After incubation, the samples were washed with elution buffer A, B, and C as in Step 1.

In the fourth step, hybridization solution C (10% (w/v) dextran sulfate, 5× SSC, 10% (v/v) formamide, 0.3% LDS, 10% (w/v) Denhardt's, and 1% VRC) was preheated to 40 °C. Then, the sample was hybridized in solution C with 100 nM signal probes for 1–3 h at 40 °C. After incubation, the samples were washed with elution buffer A, B, and C as in Step 1. Finally, the cells or tissues were stained with DAPI (1 mg/mL) for 2 min, dehydrated with gradient ethanol (70%, 85%, and 100%), sealed with an anti-fluorescence quencher, and photographed to obtain FISH images.

## Multiple round π-FISH rainbow hybridization

Multiplexed in situ hybridization and imaging were performed according to the π-FISH rainbow procedure. 21 marker genes encoding excitatory and inhibitory neurons were detected by two rounds of hybridization. In the first round, the multiplexed FISH experiments were used to simultaneously image nine excitatory neuron marker genes using different combinations of signal probes with four fluorescent molecules (Alexa Fluor 488, Alexa Fluor 546, Alexa Fluor 594, and Alexa Fluor 647). Briefly, samples were placed into the imaging buffer for image preparation following the four-step hybridization process as described in the multiplex π-FISH rainbow protocol. The imaging buffer (2× SSC, 50 mM Tris-HCl (pH 8), 10% glucose, 2 mM Trolox (Sigma-Aldrich, 238813), 0.5 mg/mL glucose oxidase (Sigma-Aldrich, G2133), and 40 µg/mL catalase (Sigma-Aldrich, C30)). After removing the fluorescent signals of the first round by 60% formamide for 1 min, the second-round hybridization in the same tissue section was performed according to the π-FISH rainbow procedure. Twelve marker genes of inhibitory neurons were detected using the multiplexed π-FISH rainbow experiment protocol. The hybridization and wash cycles were repeated as described in the first round.

## DNA in situ detection and chromosome conformation visualization by π-FISH rainbow

To delineate the chromosome conformation on the long arm of human chromosome 22, different combinations of three signal probes were used to target seven loci on the long arm, including 50 π target probes at each locus per channel. The HeLa cells were spread on the slides, followed by the same treatment as RNA in situ detection. After secondary 4% (w/v) paraformaldehyde treatment, cells were incubated with RNase A (100 mg/mL) at 37 °C for 2 h and washed three times with 2× SSC. Cells were denatured with 70% deionized formamide solution at 72 °C for 10 min and then immediately placed in cold gradient ethanol (70%, 85%, and 100%) for 1 min each. The hybridization and washing cycles were conducted as described in the multiplexed RNA π-FISH rainbow procedure, except with a 10 min wash with elution buffer A, 25% formamide treatment, and a 10 min wash with elution buffer B.

## Multiplexed in situ detection of proteins by π-FISH rainbow

Antibodies of CSFV E2 (GeneTex, Cat # GTX60997), PCV2 Cap (GeneTex, Cat # GTX128121), IFNG-Flag (Proteintech, Cat # 66008-3-Ig), Pol II (Abcam, Cat # ab5095), NeuN (Abcam, Cat # ab177487), and GABA (Invitrogen, Cat # PA5-32241) were dialyzed using a 10 KD Slide-A-Lyzer mini dialysis tube (Thermo Fisher Scientific, 69570) overnight at 4 °C.

The antibodies were concentrated at 50–100 µL with an Amicon spin filter (Sigma-Aldrich, MRCF0R030) to a concentration above 1 mg/mL. Then, the antibody solution was added with 10 times molar amount of NHS-PEG4-Azide (Thermo Fisher Scientific, 26130) and oscillated at 4°C for 2 h, followed by dialysis in PBS with stirring on ice for 4 h. After dialysis, four times the amount of DBCO-modified oligonucleotides were added and incubated with shaking at 4 °C for 12 h. Lastly, Protein A/G magnetic beads (Yepsen, 36417ES03) were used to purify oligonucleotide conjugated antibodies. The primary antibodies (CSFV E2 (1:1000 dilution), PCV2 Cap (1:1000 dilution), IFNG-Flag (1:1000 dilution), pol II (1:100, 1:1000, 1:5000, 1:10000 dilution), NeuN (1:2000 dilution), and GABA (1:2000 dilution)) were conjugated with different 47-nt ssDNA oligonucleotides.

PK-15 cells cultured in a confocal dish were transfected with AAV-IFNG-Flag plasmids. 24 h after transfection, the cells were co-infected with $1 \times 10^3$ porcine circovirus 2 (PCV2) and $1 \times 10^3$ classical swine fever virus (CSFV), and then treated with 300 mM D-Glucosamine (Sigma-Aldrich, G1514) for 1 h before culturing for 24 h to simultaneously detect IFNG, CSFV E2, PCV2 Cap, and Pol II. Cell and tissue samples were fixed with 4% (w/v) paraformaldehyde for 15 min. After dehydrating with gradient ethanol (70%, 85%, and 100%) for 5 min each, samples were subjected to 0.2 M HCl for 3 min. Cell and tissue samples were permeabilized using 0.3% (v/v) Triton X-100 for 20 min and washed with PBST for 5 min. Then, ssDNA-conjugated antibodies were incubated with the target proteins. Meanwhile, in the traditional secondary antibody control experiment, the dilution concentration of the primary antibody was performed as π-FISH rainbow and then incubated for 16 h at 4 °C. Next, the samples were hybridized with different 166 nt oligonucleotides for the next step of multiple protein detection. This hybridization was performed in solution A (consisting of 10% (w/v) dextran sulfate, 6× saline-sodium citrate (SSC), 25% (v/v) formamide, 0.3% LDS, 10% (w/v) denhardt's, and 1% VRC) for 3–4 h at 40 °C. Finally, the π-FISH rainbow probes were used for hybridization and imaging. The 47-nt ssDNA oligonucleotides and 166-nt oligonucleotides sequences (extension probes) can be found in Supplementary Data 1.

### Simultaneous in situ detection of DNA, RNA, and proteins by π-FISH rainbow

Cultured cells were fixed with 4% (w/v) paraformaldehyde for 10 min at room temperature, washed three times (5 min each) with PBS, and then dehydrated in ethanol with gradient concentrations (70%, 85%, and 100%) for 3 min each step. The cells were permeabilized using 0.3% (v/v) Triton X-100 for 30 min after treating with 0.2 M HCl for 5 min and washing three times with PBST for 5 min. Simultaneous detection of DNA, RNA, and proteins requires cell denaturation. To this end, samples were denatured in 50%–60% deionized formamide solution at 85 °C–90 °C for 10 min and then immediately transferred into ethanol with gradient concentrations (70%, 85%, and 100%) for 1 min for each step and then eluted with 2× SSC and kept at 4 °C. Next, cells were hybridized with RNA and DNA target probes, incubated with ssDNA-conjugated antibodies, and treated for complementary oligonucleotides hybridization. Finally, the secondary and tertiary amplification probes and signal probes corresponding to RNA, DNA, and protein were hybridized. The hybridization and washing cycles were conducted in the same way as the multiplexed π-FISH rainbow experimental procedures.

### In situ detection of short nucleic acid fragments by π-FISH+

To develop the π-FISH+ method, π-FISH rainbow was combined with HCR technology. The sequence information and hybridization procedures for π-FISH+ and π-FISH rainbow are the same, except that the signal probe in step 4 of π-FISH rainbow hybridization is replaced with the HCR split-probe and self-folding hairpins to further amplify the signal according to the third generation in situ hybridization chain reaction[14]. Each tertiary amplification probe can combine four HCR

amplification reactions in this process. Thus, the π-FISH+ strategy can, theoretically, use one pair of π target probes to initiate 64 HCR reactions. The HCR split probe consists of two parts (left and right). Half of the split-probe consists of three sections: bottom target region (24 nt hybridizing with the repeat regions of the tertiary amplification probe), top region (18 nt hybridizing with hairpin H1), and middle region (2 nt). The top regions of the left and right HCR split-probe constitute initiator I1 and form a sequence complementary to hairpin H1. The signal probes consist of a 72-nt hairpin H1 conjugated with fluorophores at the 3' end and a 72-nt hairpin H2 conjugated with fluorophores at the 5' end. The HCR split-probe and hairpins are synthesized DNA oligonucleotides. The protocol was conducted following steps 1 to 3 as described in the "Multiplexed RNA in situ detection by π-FISH rainbow" section. HCR were performed as previously described[14]. The probe sequences are listed in Supplementary Data 1.

### Isolation and analysis of CTCs

A total of 5 mL blood samples were collected from patients with prostate cancer and CTCs isolated using the classical Isolation by size of epithelial tumor cells (ISET) method as previously described[49]. CTCs were isolated using the AF-RCFS-11 CTC isolation machine (AnFang Biotech, Guangzhou, China). ARV7 and K19 were detected in the isolated CTCs using π-FISH+. Raw imaging data (locus 1 marked in magenta, locus 2 marked in red, and K19 (a marker of CTCs) marked in green) were obtained using confocal microscopy. Each spot (locus 1, locus 2, and K19) was automatically identified based on the different fluorescence channels using Imaris software (version 9.9). In this scenario, the specific signal corresponding to ARV7 was obtained when two fluorescence signals (locus 1 and locus 2) were colocalized. K19 gene was used as an internal control. Next, the colocalization of locus 1 and locus 2 signal spots was analyzed using Imaris software (version 9.9) and marked in yellow as ARV7 signals. ARV7 signal spots in each cell were counted.

### Construction of A549-KO cell line using CRISPR/Cas9 system

Plasmids pHKO-Cas9, pMD2.G, and pSPAX were co-transfected into HEK293T cells at a ratio of 1:1:1 for lentivirus packaging. Culture supernatant was collected 60 h after transfection, and lentivirus solution was obtained after purifying and concentrating the culture supernatant. The lentivirus solution was used to infect A549 cells, and individual positive cells were selected by adding the antibiotic blasticidin (10 µg/mL) to the culture medium and allowing the cells to proliferate to obtain the A549-Cas9 cell line. The designed sgRNAs (sgRNA 1: GTTCTCAGTGCTGGTACCAT and sgRNA 2: TCATCATCAT CAGTACACAC) were used to construct pHKO-sgRNA1-GFP and HKO-sgRNA2-GFP plasmids, respectively. pHKO-sgRNA1-GFP and pHKO-sgRNA2-GFP were then co-transfected into HEK293T cells with pMD2.G and pSPAX plasmids to obtain pHKO-sgRNA1-GFP and pHKO-sgRNA2-GFP lentivirus solutions, respectively. A549-Cas9 cells were infected by mixing the two types of lentivirus solutions. The antibiotic puromycin (2 µg/mL) was added to the culture medium to select single positive cells, which were proliferated to obtain the A549-KO cell line.

### Microscopy

Imaging for π-FISH rainbow and π-FISH+ technologies were taken using Nikon N-SIM, Nikon N-STORM, and Leica TCS SP8 STED confocal microscopy. Nikon N-SIM was used for super-resolution microscopy imaging of multiplexed genomic locus detection. Images were captured with an EMCCD camera (Andor iXon DU-897) and a 100 × 1.49 NA TIRF objective (Nikon CFI Apo TIRF). Images of multiplexed RNA π-FISH rainbow were taken using Leica TCS SP8 STED and N-STORM. Nikon NIS and Leica Elements software performed image acquisition and reconstruction.

## Image analysis

Raw imaging data obtained from Nikon N-SIM (NIS ElementsAR ver. 4.50.00), Nikon N-STORM (NIS ElementsAR ver. 5.30.02), and Leica TCS SP8 STED (LAS X ver 3.5.5.19976) instruments were processed using Imaris software (version 9.9). The analysis of the intensity surface plot and intensity profile was performed by Nikon NIS software. For multiplexed π-FISH rainbow imaging of RNA, images were subjected to a series of processes, including channel arithmetic, cell calibration, cell segmentation, RNA spot finding, assigning spots to genes, and filtering and counting genes. The list of all gene coding can be found in Source Data. The spatial structure of the long arm of chromosome 22 was analyzed using MATLAB R2018B.

## Statistics and reproducibility

Statistical tests were conducted using GraphPad Prism software (version 9). Data are presented as the mean ± s.e.m. (standard error of the mean). Two-tailed unpaired Student's $t$ test was used for comparing two groups. The statistical significance was calculated, * $P < 0.05$, ** $P < 0.01$, **** $P < 0.0001$, and ns $P > 0.05$. No data were excluded from the analyses. The experiments were not randomized. For subcellular localization patterns of *MALAT1* in HeLa cells experiments, data collection and analysis were performed by a person blinded to the treatment groups. Other experiments cannot be blinded due to objective factors, but this does not affect the conclusion of the article.

No statistical method was used to predetermine the sample size. Each experiment was replicated at least three times. For mapping the spatial cell types of mouse cerebral cortex, brain slices were acquired from at least three mice (three slices per mice).For the applications of our method in different organisms (microorganisms, plants, and animals) and distinct samples (whole-mount and tissue sections), at least 10 zebrafish whole-mount samples, 8 slices for total five maize ear, 10 brain slices from 3 mice brain, as well as tens of ASFV infected PAMs, *M. tuberculosis,* and *T. gondii* infected HFF cells were subjected for the experiments. For cell samples (HEK293T, HeLa, BHK, A549, LNCaP, PC3, PK-15), images were acquired from 20 to 60 cells for each experiment. we believe the sample size is sufficient to verify the performance of our method since all results were reliably reproduced. The precise number of cell and tissue sections is reported in the figure legends and method.

## Reporting summary

Further information on research design is available in the Nature Portfolio Reporting Summary linked to this article.

## Data availability

The authors declare that the data supporting the findings of this study are available within the article, Supplementary information, and Source data file. The sequence information of probes in this study are provided in the Supplementary information. The raw images and raw quantification data for all figures and supplementary figures are provided in source data file. Source data are provided with this paper.

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

## Acknowledgements

The authors would like to thank the State Key Laboratory of Agricultural Microbiology Core Facility for assistance. We would like to thank Yunguang Li and Chunmei Shi in Core Facility at HZAU for technical support in imaging. This work was supported by the National Natural Science Foundation of China (31941014, U21A20259 to G.C., 32171022 and 31941005 to JX.D.), the Key Research and Development Program of Guangdong Province (grant 2019B020211003 to G.C.), and the Fundamental Research Funds for the Central Universities (grant 2021YFD1800401 to G.C.). Schematics were created with BioRender.com.

## Author contributions

G.C. and JX.D. conceived the study and designed the experiments. YF.T. and XL.Z. performed all the experiments with the help of LQ.S., D.L., X.C., B.Y., and W.Z. J.Y., J.Z., and XQ.Y. assisted with the human blood samples and isolated CTCs. Z.H. provided technical assistance. F.Y. assisted with the plant studies. B.S. helped with the *T. gondii* studies. WB.Q. assisted with the ASFV studies. HY.C. helped with the data analysis. YF.T., XL.Z., G.C., and JX.D. wrote the manuscript. ZF.F. helped revise the manuscript. All authors read and approved the manuscript.

## Competing interests

The authors declare no competing interests.
