## [Peer Review File · Nature Communications]

Reviewers' Comments:

Reviewer #1:

Remarks to the Author:

Tao et al. describe a new variant of FISH, n-FISH, for the detection of RNA, DNA, or protein expression in cells or tissue. n-FISH adds additional amplification steps to increase sensitivity, and it uses a modified split probe design to reduce background. n-FISH appears to be a highly sensitive assay, and it supports multiplexed detection. The authors show impressive sensitivity, better than HCR and smFISH for an abundant target mRNA. They later show detection of less abundant target mRNAs and genomic DNA as well. The authors use n-FISH for a wide variety of applications, including cell identification in neural tissue, antibody detection, and combined detection of a microRNA and a lncRNA or a small exon, using a variant of n-FISH coupled to HCR. n-FISH appears to be a valuable and versatile addition to current molecular detection methods.

While the manuscript convincingly demonstrates several applications for the n-FISH method, there are some concerns. Important technical information for the method is not included in the manuscript (or is unclear) as discussed below. While some controls are shown in Fig. S1, additional control data to show what levels of background are present with the n-FISH assay would be valuable.

Specific comments:

For n-FISH, the current schematic in Fig. 1a does not provide enough information. There should be a more detailed explanation of the detection system in a figure, so that a reader can readily understand how it works (perhaps a more detailed schematic in a supplementary figure?). How long are the various hybridization regions and probes? How many repeats of the hybridization sequences are present? The secondary and tertiary probes are transcribed RNA, while the other sequences are DNA oligonucleotides, but that is not indicated except in the methods. There do not seem to be sequences included for the secondary or tertiary probes, although they are described as 16 and 8 repeating sequences (neither of which matches the schematic drawing in Fig 1a). Are there spacers between repeats? For n-FISH+, no description of the alternate tertiary probes (which contain HCR binding sites) is included.

For multiplex n-FISH, are individual target probes that hybridize to the different secondary probes mixed, to allow the digital encoding of the signal (as shown in S2)? Does the total number of target probes for an mRNA vary by the number of different colors used to encode the signal? Does the use of multiple probes influence background? Additional details in the methods would likely help potential users.

In Fig. 2d and 2f are the excitatory and inhibitory neuron markers labeling distinct cells as expected? Higher resolution comparison images would be helpful (like in Fig. 3g).

In Fig. 4d-f, it is hard to interpret what the two 3D chromosome structures mean without additional examples/replicates. Could the two apparent structures just be random variations or noise during fixation? Additional structure examples would help.

For Fig. 4g-h: are these individual confocal slices or stacks? 3D Reconstruction would be useful.

Fig. 5b: The target of the n-FISH+ amplification is a 54nt probe extension, which is hybridized to the miRNA, according to the methods. Can the authors show a control that the miRNA complementary sequence within the probe is necessary for the observed signal and specific? Or perhaps detect a miRNA with a different localization?

Fig. 5fg: a single patient cell is shown, without an indication of how many circulating tumor cells were scored. There is little information about the cells or analysis in the methods. A more complete experiment is needed, or the result should be omitted.

Fig. S9. HeLa cells are often reported to be triploid. Is the diploid cell possibly a failure of detection at one site? There is no indication of how many cells were scored.

Other comments:

Line 166 "low transcription" should probably be "a low level of transcripts"

Line 282 "florescent"

Fig. S8 legend should indicate what antibody/antigen is being detected (only stated in main text).

Reviewer #2:

Remarks to the Author:

This paper demonstrates pi-FISH, an advance in developing improved bi-molecular FISH probes. The intermolecular binding between probe pairs, the key advance reported in this paper, appears to substantially increase the sensitivity of bi-molecular FISH probes, without leading to a substantial increase in noise. This work, with appropriate revisions, can be broadly useful to the spatial transcriptomics community.

We do have several major comments that must be addressed before the paper can be considered for publication.

1. We could not locate the sequences of the secondary and tertiary amplifiers in the supplementary. This is essential to reproduce these results. Without these sequences, the paper cannot be considered for publication, as it will be useless to the readers.
2. Sensitivity and specificity are not convincingly demonstrated in the mouse tissue data. The pi-FISH technology clearly works for detecting high-copy number transcripts, but its ability to reliably detect low copy number transcripts in tissue samples, and minimize technical noise are not thoroughly demonstrated.
3. The expected long length of pi-FISH amplifiers may limit their diffusion in tissue – these concerns could be alleviated by a single-molecule colocalization experiment with another validated high-sensitivity FISH method.
4. Related to this comment, for the mouse brain data, where there is ample published data from other methods (such as osmFISH and MERFISH), a gene by gene and a cell annotation comparison is required to show the performance of pi-FISH
5. It would be best to swap figure S1B with 1C. S1B better represents what an actual experiment is expected to look like. Figure 1C looks like a best-case scenario and cannot be supported as a representative sample without more data on other probe pairs.

Reviewer #3:

Remarks to the Author:

Tao et al., present a split-FISH probe design coupled with branched amplification and later hybridization chain reaction in a variety of model systems, with the overall goal of enabling the detection of short RNAs. In general, this is of broad interest in the spatial-transcriptomics field. The authors go on to present different modalities of this FISH technique to investigate DNA chromosome conformation and indicate antibody oligo-conjugation protocols are possible. Overall, the techniques presented here are not very novel as they are highly similar to other FISH signal amplification strategies (SABER, branch DNA amplification, split-FISH). Moreover, multiplexed imaging of DNA, RNA and protein (e.g. INCITE-seq, ORCA, MINA, seqFISH, Slide-seq etc.) are routinely performed by other groups and are much higher-throughput compared to those presented in this manuscript (i.e. simultaneous imaging of 100s to 10,000s of RNA, DNA loci and dozens of proteins in tissue). As well, a significant biological question or finding was not addressed in this paper. Therefore, I recommend that this paper should not be published in Nature Communications.

My recommendation to the authors would be to look at other papers recently published on FISH signal amplification or multiplexed-omics technologies for guidance on a future manuscript

(Multiplexed Detection of RNA using MERFISH and branch amplification- Xia et al., 2019). In my opinion, the manuscript suffers greatly from a general lack of quantification and necessary controls that would be required to support the authors claims of enhanced specificity, sensitivity and efficiency of this FISH method. I found the manuscript would benefit from the inclusion of much more detail as this is a technical paper- a detailed methods section is lacking in this manuscript. A single π -probe for signal amplification is a desirable design in the method presented here, but there is insufficient data to currently support the authors claims as described in more detail below:

The authors claim this method enhances signal intensity of individual FISH foci in varying modalities (1E,1G,3D-E, S1E). The authors measure signal intensity by taking a line average across the entire cell (S3, S7, S8). This does not in fact quantify signal intensity of individual foci, but rather measures the intensity across the specified area, which is simply brighter compared to controls because there are increased numbers of fish foci as shown by their results in Figure 1B. With this, the entire claim of enhanced signal intensity cannot be concluded.

Figure 1B/S1D. What is the specificity of the primary π -probe binding to the target mRNA? What is the specificity of the tertiary and readout probes binding to secondary π -probes and to each other?

Figure S1D shows you can have non-specific, off target interactions between two unilateral π -fish probes with 0 complimentary bases. This suggests this method does not have high specificity. What are the off-target rates and thus, false positive rates of detection using this method? What about these rates when multiple RNA species are analyzed simultaneously during multiplexing later in the manuscript?

More generally, what is the signal intensity variability and detection efficiency of the π -probes methods?

Figure 1D/S1E. Does increasing the number of π -FISH for a given mRNA alter the FISH foci size? Moreover, what is the size of a single FISH foci using only one π -probe? Increased foci size can result in the cytoplasmic crowding of RNA FISH signal preventing proper identification/segmentation of individual foci during analysis i.e Fig1E- π -probe condition. This is undesirable as this will impair proper mRNA copy number counts- see Xia et al., 2019 branch DNA amplification paper.

Figure 1E. smFISH-FL is reported to give the most accurate mRNA counts compared to RNA-sequencing methods. I am therefore somewhat surprised and confused that both HCR and π -FISH yield almost double the RNA counts per cell of smFISH-FL method? Can the authors compare their counts to what others have found? As well, this may indicate off-target effects.

Figure 2-3

Can the authors explain what channel arithmetic entails as a decoding scheme and present its misidentification rates and detection efficiency? Is there an error-robust correction strategy the authors could use?

Refer to above about issues surrounding the quantitation method for signal intensity.

How many π -probes are used per ssDNA oligo conjugated antibody? What are the off-target, false-positive and misidentification rates for all the oligos presented during this design?

Figure 4

S9. To determine the specificity of the DNA π -probes to centromeres, the authors need to co-localize the π -probe signal with a centromere marker.

Figure 5

In the opening sentence the authors claim 1KB is required for current FISH methods which is not accurate. 500bp for mRNA has been achieved; see Xia et al., 2019- Spatial transcriptome profiling by MERFISH

Similar to concerns presented in Figure 1, the introduction of HCR here requires the presentation

of appropriate controls. How was HCR tunability achieved, number of repetitive elements following HCR that were used in final probe design, off target rates and specificity of readout probes to HCR repetitive elements, information about the self-folding hairpin sequence, signal intensity fold changes associated with HCR, signal intensity variation, detection efficiency, foci size etc

The above controls also apply to the 54bp extension probe for miR-145.

Figure 5B Malat1 staining looks dramatically different than controls shown in S10ab. Thus, this cannot be used to support the efficiency and accuracy of this method.

Figure 5G; needs controls outlined above

For the CRIPSR experiments, more evidence/descriptions of these mutations and methodology needs to be presented. Moreover, this method only works if mutations are very large and not small as the authors have suggested given they are investigating a 2KB deletion.

Reviewer #4:

None

Reviewers' Comments:

Reviewer #1 (Remarks to the Author):

Tao et al. describe a new variant of FISH, π -FISH, for the detection of RNA, DNA, or protein expression in cells or tissue. π -FISH adds additional amplification steps to increase sensitivity, and it uses a modified split probe design to reduce background. π -FISH appears to be a highly sensitive assay, and it supports multiplexed detection. The authors show impressive sensitivity, better than HCR and smFISH for an abundant target mRNA. They later show detection of less abundant target mRNAs and genomic DNA as well. The authors use π -FISH for a wide variety of applications, including cell identification in neural tissue, antibody detection, and combined detection of a microRNA and a lncRNA or a small exon, using a variant of π -FISH coupled to HCR. π -FISH appears to be a valuable and versatile addition to current molecular detection methods.

Response: Thank you for your comments and valuable suggestions. We have conducted additional experiments and included further supporting data based on your suggestions. All changes have been tracked with yellow highlights in the revised manuscript. Please see our point-by-point responses to your comments below.

While the manuscript convincingly demonstrates several applications for the π -FISH method, there are some concerns. Important technical information for the method is not included in the manuscript (or is unclear) as discussed below. While some controls are shown in Fig. S1, additional control data to show what levels of background are present with the π -FISH assay would be valuable.

Response: Thank you for your valuable comments. As suggested, we have added the technical details on the method in the revised manuscript. We also performed additional control experiments in cells and tissues to analyse the background signals generated using π -FISH rainbow.

As shown in Fig. R1-1a-d, we added negative controls, including unilateral target probes, no target probes, RNase treatment, and probes targeting the bacterial gene *dapB* during the detection of *ACTB*, *PPIA*, and *B2M* genes in HeLa cells (Fig. R1-1a and b), as well as *Ctgf*, *Penk*, and *Sst* genes in mouse brain tissues (Fig. R1-1c and d). We used a series of gradient elution buffers to wash the samples after each hybridization step, minimizing noise and keep the positive signals. As shown in Fig. R1-1e-h, there were negligible signal spots in the control cells and tissues, indicating that π -FISH rainbow

is associated with high specificity and low background noise of π -FISH rainbow. We have added this figure as Supplementary Fig. 4 in the revised manuscript.

Fig. R1-1. Background of π -FISH rainbow in cells and tissues.

(a-b) The background of π -FISH rainbow in cells was verified by *in situ* detection of *ACTB* (a) and co-detection of *ACTB*, *PPIA*, and *B2M* (b) along with multiple negative controls, including unilateral target probes, no target probes, bilateral target probes with RNase treatment, and probes targeting the bacterial gene *dapB* in HeLa cells, respectively. Scale bars, 10 μ m.

(c-d) The background of π -FISH rainbow in tissues was verified by *in situ* detection of *Ctgf* (c) and co-detection of *Ctgf*, *Penk*, and *Sst* (d) along with multiple negative controls, including unilateral target probes, no target probe, bilateral target probes with RNase treatment, and probes targeting the bacterial gene *dapB* in mouse brain sections, respectively. (i-v) Higher magnification of square regions in the upper panels of (c) and (d) were shown in the lower panels. Scale bars, 100 μ m (top) and 20 μ m (bottom).

(e-h) The histogram of spot counts per cell for positive signals and background noise from (a-d). n = 30 cells per group.

Specific comments:

For π -FISH, the current schematic in Fig.1a does not provide enough information. There should be a more detailed explanation of the detection system in a figure, so that a reader can readily understand how it works (perhaps a more detailed schematic in a supplementary figure?). How long are the various hybridization regions and probes? How many repeats of the hybridization sequences are present? The secondary and tertiary probes are transcribed RNA, while the other sequences are DNA oligonucleotides, but that is not indicated except in the methods. There do not seem to be sequences included for the secondary or tertiary probes, although they are described as 16 and 8 repeating sequences (neither of which matches the schematic drawing in Fig 1a). Are there spacers between repeats? For π -FISH+, no description of the alternate tertiary probes (which contain HCR binding sites) is included.

Response: Thank you for this comment. We have added detailed schematics of the π -FISH rainbow and π -FISH+ procedures in the revised manuscript (Supplementary Fig. 1 and 13). The schematics include key information on the probe lengths, hybridization regions, hybridization sequence repeats, complementary base pairing, and spacers. We have added the sequences of the secondary and tertiary probes to Supplementary Table 1.

Fig. R1-2 shows a detailed workflow for π -FISH rainbow (presented as Supplementary Fig. 1 in the revised manuscript). The π -FISH rainbow amplification system consists of four hybridization steps. Step 1: Hybridization of π target probes with the target sequence. The π target probes contain 2–4 complementary base pairs in

the middle region, which form a π shape. Half of the π target probe consists of three sections: bottom target region (20–25 nucleotides (nt) for hybridization with target sequence), top region (14 nt for hybridization with secondary probes), and middle region (8 nt, 2–4 nt of which are designed to form bonds between π target probe pairs).

Step 2: Hybridization of secondary amplification probes with π target probes. The secondary U-shaped amplification probes (509 nt) consist of two sections: the middle region and the 5' and 3' arm regions. The middle region (29 nt) hybridizes with the top region from the left and right sides of the π target probe pairs. The 5' and 3' arm regions consist of 16 repeating sequences, with each repeat containing a 10 nt spacer region and a 20 nt hybridization region for binding the tertiary probes.

Step 3: Hybridization of tertiary amplification probes with secondary amplification probes. The tertiary U-shaped amplification probes (260 nt) consist of two sections: the middle region and the 5' and 3' arm regions. The middle region (20 nt) is designed to hybridize with the repeat regions of the 5' and 3' arms of the secondary amplification probes. The 5' and 3' arm regions consist of 8 repeating sequences, each containing a 10 nt spacer region and a 20 nt hybridization region for binding the signal probes.

Step 4: Hybridization of signal probes with tertiary amplification probes. The 20 nt signal probe is conjugated with fluorophores at both 5' and 3' ends and is then used to hybridize with repeat regions of the tertiary amplification probes. As described in the “probe synthesis and preparation” subsection of the methods section of the revised manuscript, π target probes and signal probes are synthesized DNA oligonucleotides. Secondary and tertiary amplification probes are ssDNA probes generated from plasmid templates through *in vitro* transcription and reverse transcription.

Fig. R1-2. The workflow of π -FISH rainbow and the probe sequence information.

Step 1: Hybridization of π target probes with the target sequence. The π target probe contains 2–4

complementary base pairs in the middle bond region to form a π shape. Half of the π target probe consists of three sections: bottom target region (20–25 nt), top region (14 nt), and middle region (8 nt). Steps 2 and 3: Hybridization of amplification probes. The total lengths of secondary and tertiary amplification probes are 509 nt and 260 nt, respectively. And both of them consist of two sections: the middle region and the 5' and 3' arm regions. Step 4: Hybridization of signal probes with amplification probes. The signal probe is 20 nt and conjugated with fluorophores at both 5' and 3' ends. All probe sequences are listed in Supplementary Table 1.

The sequence information and hybridization procedures for π -FISH+ and π -FISH rainbow are the same, except that the signal probe in step 4 of π -FISH rainbow is replaced with the HCR split-probe and self-folding hairpins to further amplify the signal according to the third generation *in situ* hybridization chain reaction¹ (as shown in Fig. R1-3). Each tertiary amplification probe can combine four HCR split-probes to initiate four HCR amplification reactions in this process. The HCR split-probe consists of the left and right parts. Half of the split-probe consists of three sections: bottom target region (24 nt, hybridizing with the repeated regions of the tertiary amplification probe), top region (18 nt, hybridizing with hairpin H1), and middle region (2 nt). The top regions of the left and right sides of the HCR split-probe constitute initiator I1 and form a sequence complementary to hairpin H1. The signal probes consist of 72 nt hairpin H1 conjugated with fluorophores at the 3' end and 72 nt hairpin H2 conjugated with fluorophores at the 5' end. The HCR split-probe and hairpins are synthesized DNA oligonucleotides. We have added this schematic as Supplementary Fig. 13 in the revised manuscript.

Fig. R1-3. Schematic overview and probe sequence information for π -FISH+.

For π -FISH+, the sequence information and hybridization procedure of steps 1–3 are the same as in

π -FISH rainbow, while the signal probe in step 4 of π -FISH rainbow is replaced with the HCR split-probe and self-folding hairpins to further amplify the signal according to HCR 3.0. Each tertiary amplification probe can combine four HCR split-probes to initiate four HCR amplification reactions in this process. The HCR split-probe consists of left and right parts. Half of the split-probe consists of three sections: bottom target region (24 nt), top region (18 nt), and middle region (2 nt). The top regions on the left and right sides of the HCR split-probe constitute initiator I1 and form a complementary sequence to hairpin H1. The signal probes consist of hairpin H1 and H2 with 72 nt each.

For multiplex π -FISH, are individual target probes that hybridize to the different secondary probes mixed, to allow the digital encoding of the signal (as shown in S2)? Does the total number of target probes for an mRNA vary by the number of different colors used to encode the signal? Does the use of multiple probes influence background? Additional details in the methods would likely help potential users.

Response: We apologize for the unclear description. To generate multiplex signals for one RNA, we divided π target probes into different groups. Each group's π target probes are hybridized with the same secondary probes. Combining of different groups of probes allows amplification of the signal encoding the multiplex signal. Theoretically, combining of four fluorescent signal probes can generate 15 ($C_4^1 + C_4^2 + C_4^3 + C_4^4$) signal codes and decode 15 genes in one go (Fig. R1-4; Supplementary Fig. 3a in the revised manuscript). The number of π target probes per mRNA increases based on the number of fluorescent colours, producing a consistent luminance for each fluorescence signal. As suggested, we have added these details to the methods section of the revised manuscript.

Schematic of signal encoding of multiplex π -FISH rainbow

Fig. R1-4. Strategies for multiplexed detection of π -FISH rainbow.

Schematic of signal coding based on the single or merged color of double, triple, and quadruple fluorescence channel signal for multiplexed π -FISH rainbow. Thus, four fluorescence channels will generate four coding for single channel signal (C_4^1), six coding for the merged signal of double channels (C_4^2), four coding for the merged signal of triple channels (C_4^3), and one coding for the merged single of four channels (C_4^4).

To evaluate the background of multiplex probes, we conducted multichannel detection of *GadI* and *Vip* genes in mouse brain tissues using multiple controls, including bilateral target probes, unilateral target probes, no target probes, RNase treatment, and bacterial *dapB* probes. As shown in Fig. R1-5, almost no background was observed in the multichannel detection experiment, indicating that multiplex probes have no influence on the background.

Fig. R1-5. Background of multichannel detections of *Gad1* and *Vip* genes by π -FISH rainbow. (a-d) *In situ* detection of *Gad1* and *Vip* mRNA with three fluorescence channels. The background of π -FISH rainbow was verified with multiple controls, including unilateral target probes, no target probes, RNase treatment, and bacterial *dapB* probes in three fluorescence channels for the hybridization in mouse brain sections, respectively. Magnified images showed the specificity and background noise of π -FISH rainbow (left bottom). Ch 1, channel 1. Ch 2, channel 2. Ch 3, channel 3. Scale bars, 100 μ m (lower magnification) and 10 μ m (high magnification).

In Fig. 2d and 2f are the excitatory and inhibitory neuron markers labeling distinct cells as expected? Higher resolution comparison images would be helpful (like in Fig. 3g).

Response: Yes, the excitatory and inhibitory neuron markers labelled distinct cells as expected. As suggested, we have provided higher resolution images in Fig. R1-6

(Supplementary Fig. 9c in the revised manuscript). Based on the marker genes of excitatory and inhibitory neurons, cells in the S1 region of the mouse brain clustered into three major types: excitatory neurons (orange), inhibitory neurons (green), and non-neuronal cells (gray). Statistics based on three biological replicates revealed that the proportions of excitatory, inhibitory, and non-neuronal cells in the S1 cortex were 51%, 12%, and 37%, respectively. These results are consistent with those of previous studies^{2,3}.

Fig. R1-6. Representative cell type-resolved spatial map in mouse primary somatosensory cortex (S1).

Cells of the S1 cortex were marked as three different types: excitatory neurons (orange, yellow arrow indicated), inhibitory neurons (green, green arrow indicated), and non-neuronal cells (gray, white arrow indicated). Higher-resolution comparison images of the cells were listed at the bottom. Scale bars, 100 μm (top) and 5 μm (bottom).

In Fig. 4d-f, it is hard to interpret what the two 3D chromosome structures mean without additional examples/replicates. Could the two apparent structures just be random variations or noise during fixation? Additional structure examples would help.

Response: Thank you for the comment. As suggested, we have included five additional replicates to explore the 3D spatial conformations of the long arm of homologous chromosome 22. As shown in Fig. R1-7, all the cells contain two homologous chromosomes with relatively distinct 3D structures. We extensively explored published literature in this field and found that previous studies observed phenomena consistent with our data^{4,5}. Thus, we believe the two apparent structures are not random variations or noise occurring during fixation.

Fig. R1-7. 3D image and spatial conformation of the homologous chromosome structure in individual cells.

The hybridization signals for seven loci (rectangular box i and ii) on the long arm of chromosome 22 were decoded and reconstructed in the 3D image (top). The spatial conformation of the homologous chromosome structures of the long arm of chromosome 22 based on hybridization signals was decoded (bottom). Scale bars, 3 μm .

For Fig. 4g-h: are these individual confocal slices or stacks? 3D Reconstruction would be useful.

Response: The images in Fig. 4g and h in the original manuscript were individual confocal slices. As suggested, we included additional 3D reconstruction data from confocal Z-axis scanning to show the spatial distribution of DNA, RNA, and proteins (Fig. R1-8). These high-quality FISH signals of different biomolecules suggest the broad potential application of π -FISH for multi-omics investigation. We have replaced the previous slice images with 3D reconstructions in revised Fig. 4g and h.

Fig. R1-8. Simultaneous *in situ* detection of DNA, RNA, and protein by π -FISH rainbow.

(a) Simultaneous detection of lncRNA *NEAT1* and its target *MAPK15* genomic locus in HeLa cells was imaged by confocal Z-axis scanning. *NEAT1* and *MAPK15* signal probes were labeled with Alexa Fluor 647 (magenta) and Alexa Fluor 488 (green), respectively. Scale bars, 2.5 μm .

(b) Simultaneous detection of protein (Pol II protein, red), RNA (*NEAT1* transcript, magenta), and its target genomic DNA locus (*MAPK15*, green) using π -FISH rainbow in HeLa cells followed by confocal Z-axis scanning. Scale bars, 2.5 μ m.

Fig. 5b: The target of the π -FISH+ amplification is a 54nt probe extension, which is hybridized to the miRNA, according to the methods. Can the authors show a control that the miRNA complementary sequence within the probe is necessary for the observed signal and specific? Or perhaps detect a miRNA with a different localization?

Response: Thank you for the comment. *miR145-5p* only has a 23-bp seed sequence that is insufficient for hybridizing the π target probe. Therefore, the extension probe is required to provide sufficient space for the π -FISH+ amplification system. To verify the specificity of this extension probe, we analysed additional controls, including scramble extension probe (sequence at the miRNA complementary region is scrambled) and no extension probe. As shown in Fig. R1-9a-c, there is almost no noise signal in control cells, indicating that the ssDNA extension probe is specific and necessary for miRNA detection. We have added this figure as Supplementary Fig. 15 in our revised manuscript.

Fig. R1-9. Validation of the specificity and efficiency of π -FISH+ for miRNA detection.

(a) The *miR145-5p* was detected using a specific extension probe as a positive control. Scale bar, 2.5 μ m.

(b-c) The specificity of the miRNA complementary sequence within the extension probe was verified with a scramble extension probe (nonspecific binding, b) and no extension probe (c) for the hybridization. Scale bars, 2.5 μ m.

Fig. 5fg: a single patient cell is shown, without an indication of how many circulating tumor cells were scored. There is little information about the cells or analysis in the methods. A more complete experiment is needed, or the result should be omitted.

Response: Thank you for the suggestion. We originally analysed 20 circulating tumour cells (CTCs) from the blood of patient with prostate cancer. To verify these results, we collected blood from two more patients with prostate cancer and analysed 15 and 18

CTCs from each patient. As shown in Fig. R1-10, compared with healthy human blood cells, the *ARV7* splice variant was successfully detected in CTCs from prostate cancer patients, indicating the integrity of this strategy as an anti-androgen therapy resistance diagnosis. We have added a detailed description of the isolation and analysis of CTCs in the revised manuscript (lines 771 to 784). We have added Fig. R1-10c-e as Supplementary Fig. 17 in the revised manuscript.

Fig. R1-10. Detection of *ARV7* splice variant in circulating tumor cells from prostate cancer patients.

(a) Schematic for circulating tumor cells (CTCs) isolation from prostate cancer patients and *ARV7* detection by π -FISH+.

(b) Diagram of probe design for identifying the *ARV7* variant by simultaneous detection of locus 1 (exist in *ARV1*, *ARV2*, *ARV4*, and *ARV7*) and locus 2 (exist in *ARV5* and *ARV7*). The *K19* gene was used as an internal control.

(c) Detection and decoding of *K19*, *ARV7* locus 1, and *ARV7* locus 2. The co-localization of the

locus 1 and 2 signals was identified as *ARV7* (right). Scale bars, 5 μm .

(d) Application of π -FISH+ for *ARV7* detection in multiple CTCs from two more prostate cancer patients. Scale bars, 5 μm .

(e) Healthy human blood cells were used as negative controls. Scale bars, 5 μm .

Fig. S9. HeLa cells are often reported to be triploid. Is the diploid cell possibly a failure of detection at one site? There is no indication of how many cells were scored.

Response: Thank you for this important comment. Previous studies have indeed reported that HeLa cells can exist as diploids or triploids⁶. To verify this, we scanned 89 HeLa cells and observed that 29.2% (26/89) were diploid while 47.2% (42/89) were triploid (Fig. R1-11), consistent with the results of previous studies. Notably, almost no haploid cells were observed, suggesting the high efficiency of π -FISH rainbow for DNA detection. We have added this figure as Supplementary Fig. 12b-c in the revised manuscript.

Fig. R1-11. Detection of chromosome copy number in HeLa cells.

(a-b) Diagram (left) and detection (right) of diploid (a) and triploid (b) chromosome 2 in HeLa cells. Scale bars, 2.5 μm .

Other comments:

Line 166 “low transcription” should probably be “a low level of transcripts”

Response: As suggested, we changed the expression “low transcription” to “a low level of transcripts” in the revised manuscript (lines 189).

Line 282 “florescent”

Response: We apologize for this mistake and appreciate the reviewer’s careful checking. We have corrected this typing error in the revised manuscript (lines 311).

Fig. S8 legend should indicate what antibody/antigen is being detected (only stated in main text).

Response: As suggested, we have added information on the antibody in the legend of Supplementary Fig. 10 in the revised manuscript.

Reviewer #2 (Remarks to the Author):

This paper demonstrates pi-FISH, an advance in developing improved bi-molecular FISH probes. The intermolecular binding between probe pairs, the key advance reported in this paper, appears to substantially increase the sensitivity of bi-molecular FISH probes, without leading to a substantial increase in noise. This work, with appropriate revisions, can be broadly useful to the spatial transcriptomics community. We do have several major comments that must be addressed before the paper can be considered for publication.

Response: Thank you for your positive feedback. We have made appropriate revisions based on your comments and suggestions. All changes have been tracked with yellow highlights in the revised manuscript. Please see our point-by-point responses to your comments below.

1. We could not locate the sequences of the secondary and tertiary amplifiers in the supplementary. This is essential to reproduce these results. Without these sequences, the paper cannot be considered for publication, as it will be useless to the readers.

Response: Thank you for this important comment. We have added the sequences of the secondary and tertiary amplification probes to Supplementary Table 1 in the revised manuscript. We have also provided a detailed schematic of the π -FISH rainbow system (shown in Fig. R2-1) as Supplementary Fig. 1 in the revised manuscript.

Fig. R2-1. The workflow of π -FISH rainbow and the probe sequence information.

Step 1: Hybridization of π target probes with the target sequence. The π target probe is designed

with 2–4 complementary base pairs in the middle bond region to form a π shape. Half of the π target probe consists of three sections: bottom target region (20–25 nt), top region (14 nt), and middle region (8 nt). Steps 2 and 3: Hybridization of amplification probes. The total lengths of secondary and tertiary amplification probes are 509 nt and 260 nt, respectively. And both of them consist of two sections: the middle region and the 5' and 3' arm regions. Step 4: Hybridization of signal probes with amplification probes. The signal probe is 20 nt and conjugated with fluorophores at both 5' and 3' ends. All the sequences are listed in Supplementary Table 1.

2. Sensitivity and specificity are not convincingly demonstrated in the mouse tissue data. The pi-FISH technology clearly works for detecting high-copy number transcripts, but its ability to reliably detect low copy number transcripts in tissue samples, and minimize technical noise are not thoroughly demonstrated.

Response: Thank you for this comment. We originally detected *Cux2* and *Pcp4* genes in mouse brain tissues (data presented in Supplementary Fig. 4a and b in the original manuscript and in Fig. R2-2a and b here). As suggested, to verify the sensitivity of π -FISH rainbow in tissue samples, we attempted to detect the expression of *Lrmp* and *Ptpru* genes, which have low abundance transcripts, based on ISH results from the Allen Institute of Brain Science⁷ (Fig. R2-2c and d). Our data showed that π -FISH rainbow faithfully reproduced the spatial distributions of both high (*Cux2* and *Pcp4*) and low (*Lrmp* and *Ptpru*) abundant expression genes, indicating the sensitivity and reliability of π -FISH rainbow to analyse samples. We have added this figure as Supplementary Fig. 7a-d in the revised manuscript.

Fig. R2-2. Validation of highly and lowly expressed genes by π -FISH rainbow.

(a-d) The expression patterns of highly expressed genes *Cux2* (a) and *Pcp4* (b) and lowly expressed genes *Lrmp* (c) and *Ptpru* (d) in mouse brain detected by π -FISH rainbow were consistent with that described by the Allen Institute of Brain Science (AIBS). Scale bars, 50 μ m (a and b); 100 μ m (c and d).

To assess the noise level of π -FISH rainbow, we detected *Ctgf*, *Penk*, and *Sst* genes in mouse brain tissues using different negative controls, including unilateral target probes, no target probes, RNase treatment, and bacterial *dapB* probes. We used a series of gradient elution buffers to wash the samples to minimize noise and keep the positive signal. As shown in Fig. R2-3a and d, there was almost no signal spot in the control groups. This high specificity and low background noise of π -FISH rainbow is most likely due to the stability of our π -FISH hybridization complex, which increases resistance to the wash steps. We have added this information as Supplementary Fig. 4c, d, g, and h in our revised manuscript.

Fig. R2-3. Verification of background noise of π -FISH rainbow in tissues.

(a-b) The background of π -FISH rainbow in tissues was verified by *in situ* detection of *Ctgf* (a) and co-detection of *Ctgf*, *Penk*, and *Sst* (b) along with multiple negative controls, including unilateral target probes, no target probe, bilateral target probes with RNase treatment, and probes targeting the bacterial gene *dapB* in mouse brain sections, respectively. (i-v) Higher magnification of square regions in the upper panels of (a) and (b) were shown in the lower panels. Scale bars, 100 μ m (top) and 20 μ m (bottom).

(c-d) The histogram of spot counts per cell for positive signals and background noise from (a-b). $n = 30$ cells per group.

To further verify the specificity and background noise of π -FISH rainbow in tissue

samples, we examined the mutually exclusive expression of different marker genes in mouse brain. *Slc17a7* and *Gad1* are excitatory and inhibitory neuron markers, respectively⁸. *Vip* and *Sst* gene expression label mutually-exclusive subclasses of interneurons in the cerebral cortex⁹. As shown in Fig. R2-4, our results showed that these marker genes had distinct expression patterns. Higher-resolution images showed almost no *Gad1* expression (green) in *Slc17a7*⁺ neurons (red). Likewise, there was scarcely any *Slc17a7* expression in *Gad1*⁺ neurons. We also observed the same mutually exclusive expression pattern for *Vip* and *Sst*. These results demonstrate the specificity and accuracy of π -FISH rainbow. We have added this figure as Supplementary Fig. 7e and f in our revised manuscript.

Fig. R2-4. The specificity of π -FISH rainbow was verified by detecting the mutually exclusive expression of different neuron markers.

(a) Co-detection of *Sst* and *Vip*, two mutually exclusively expressed genes in different subclasses of interneurons in the mouse cerebral cortex, by π -FISH rainbow. The green and red arrows indicated *Sst*⁺ and *Vip*⁺ neurons, respectively. Scale bars, 100 μ m.

(b) Co-detection of inhibitory (*Gad1*) and excitatory (*Slc17a7*) neuron markers in mouse cerebral cortex by π -FISH rainbow. The green and red arrows indicated *Gad1*⁺ and *Slc17a7*⁺ neurons, respectively. Scale bars, 100 μ m.

3. The expected long length of pi-FISH amplifiers may limit their diffusion in tissue – these concerns could be alleviated by a single-molecule colocalization experiment with

another validated high-sensitivity FISH method.

Response: Thank you for the comment. We measured the expression of *Ctgf* and *Gad1* genes in mouse brain tissue using highly sensitive hybridization chain reaction (HCR)-FISH and π -FISH rainbow for single-molecule co-localization to measure the diffusion of π -FISH amplifiers. As shown in Fig. R2-5a and b, full co-localization between two fluorescent signals was observed, indicating efficient probe diffusion and signal amplification ability of π -FISH rainbow. The consistency in hybridization results between conventional ISH (Fig. R2-2) and HCR-FISH confirm that the diffusion of π -FISH amplifier probes in the tissue was effective.

Fig. R2-5. Simultaneous detection of *Ctgf* and *Gad1* by π -FISH rainbow and HCR.

(a-b) *Ctgf* (a) and *Gad1* (b) were detected in the mouse brain tissues by both π -FISH rainbow and HCR, respectively. White boxes represented the higher-resolution images. The red and green arrows indicated π -FISH rainbow and HCR signals, respectively. Scale bars, 100 μ m.

4. Related to this comment, for the mouse brain data, where there is ample published data from other methods (such as osmFISH and MERFISH), a gene by gene and a cell annotation comparison is required to show the performance of pi-FISH

Response: Thank you for this comment. We analysed mouse cerebral cortex gene expression data using π -FISH rainbow and compared the results with published data from osmFISH² and MERFISH¹⁰, gene-by-gene. Since there are no publicly available MERFISH generated mouse somatosensory cortex data, we used the mouse primary motor cortex (MOp) MERFISH data for comparison. As shown in Fig. R2-6, we

compared the expression patterns of the genes (*Rorb*, *Cux2*, *Sulf2*, *Foxp2*, and *Cxcl14*), which have been investigated using both π -FISH rainbow and MERFISH. The similar patterns of expression of these genes in the mouse cerebral cortex detected using both methods indicate the reproducibility of π -FISH rainbow. As shown in Fig. R2-7, we also compared π -FISH rainbow data with osmFISH data from the mouse somatosensory cortex and observed that the expression patterns of genes in the two datasets (*Rorb*, *Vip*, *Crh*, and *Cpne5*) were highly consistent, further indicating the reliable performance of π -FISH rainbow.

Fig. R2-6. Comparison of π -FISH rainbow and MERFISH performance.

(a-j) Comparing the expression patterns of genes (*Rorb*, *Cux2*, *Sulf2*, *Foxp2*, and *Cxcl14*) analysed using π -FISH rainbow data from mice somatosensory cortex and MERFISH data from mice primary motor cortex (MOp).

Fig. R2-7. Comparison of π -FISH rainbow and osmFISH performance.

(a-h) Comparing the expression patterns of genes (*Rorb*, *Vip*, *Crh*, and *Cpne5*) analysed using π -FISH rainbow data and osmFISH data both from the somatosensory cortex.

. It would be best to swap figure S1B with 1C. S1B better represents what an actual experiment is expected to look like. Figure 1C looks like a best-case scenario and cannot be supported as a representative sample without more data on other probe pairs.

Response: As suggested, we have swapped Supplementary Fig. 1b with Fig. 1c in the revised manuscript.

Reviewer #3 (Remarks to the Author):

Tao et al., present a split-FISH probe design coupled with branched amplification and later hybridization chain reaction in a variety of model systems, with the overall goal of enabling the detection of short RNAs. In general, this is of broad interest in the spatial-transcriptomics field. The authors go on to present different modalities of this FISH technique to investigate DNA chromosome conformation and indicate antibody oligo-conjugation protocols are possible. Overall, the techniques presented here are not very novel as they are highly similar to other FISH signal amplification strategies (SABER, branch DNA amplification, split-FISH). Moreover, multiplexed imaging of DNA, RNA and protein (e.g. INCITE-seq, ORCA, MINA, seqFISH, Slide-seq etc.) are routinely performed by other groups and are much higher-throughput compared to those presented in this manuscript (i.e. simultaneous imaging of 100s to 10,000s of RNA, DNA loci and dozens of proteins in tissue). As well, a significant biological question or finding was not addressed in this paper. Therefore, I recommend that this paper should not be published in Nature Communications.

Response: Thank you for the feedback and comments. We agree with the reviewer that recent progress in FISH methods such as SABER, branch DNA amplification, split-FISH, SeqFISH, and MERFISH has significantly improved the intensity and throughput of FISH. However, the new generation of FISH technologies is still in the infancy stage and there are various unresolved challenges that need to be addressed, including (1) detection of short nucleic acid sequences (e.g., microRNA), specific splicing variants, and narrow genomic loci; (2) simultaneous triple detection of DNA, RNA, and proteins for *in situ* multi-omics investigations such as regulatory bio-macromolecular complexes, composed of non-coding RNA, the corresponding target DNA, and the associated proteins; (3) high signal intensity and high efficiency while concurrently maintaining low background noise; (4) While SeqFISH and MERFISH were huge breakthroughs in high throughput FISH and greatly contributed to spatial omics, they require very complex bioinformatic analysis and special equipment for multiple-round hybridizations and are highly technically demanding, which are still big challenges for many laboratories.

In this regard, π -FISH rainbow has considerable advances, novelties, and superiorities: (1) We optimized the hybridization protocol and realized simultaneous detection of DNA, RNA, and protein. Although other groups perform multiplexed imaging of these biomolecules, simultaneous triple detection of DNA, RNA, and protein has not been reported. DNA, RNA, and protein co-detection will provide much

more detail for gene expression regulation and multi-omics investigation. (2) Designing π target probe pairs with middle bonds increase the stability of the hybridization complex. This innovation increases both hybridization specificity and efficiency. (3) π -FISH can detect short-length targets, including miRNA, which is extremely challenging to do with conventional FISH, smFISH, split-FISH, and seqFISH methods. (4) By combining different fluorescent probes, π -FISH can decode the spatial profiles of 21 genes at single-cell resolution using only two rounds of hybridization. This relatively simple method does not need complex bioinformatic analysis and can be carried out using routine equipment. We believe this simple and robust method can be used extensively in many molecular biology research laboratories, especially for post-single-cell sequencing research.

Moreover, we employed π -FISH rainbow for several important applications and obtained several new biological findings: (1) We designed a set of probes and successfully detected the androgen receptor splice variant 7 (*ARV7*) in circulating tumour cells (CTCs). Distinguishing *ARV7* from other splice variants is desperately needed for therapy-resistance diagnosis in prostate cancer; (2) With the integrated analysis of single-cell sequencing data, we delineated the spatial landscape of 13 subclasses of inhibitory neurons in mice S1 cortex. We found that Int 9 and Int 10 subclasses were mainly distributed in the superficial layers (L2/L3), whereas Int 6 subclass was predominately located in the deep layers (L5 to L6); (3) Owing to the high sensitivity of π -FISH, we observed for the first time that *MALAT1*, a key regulator of tumour cell cycle¹¹, has three distinct subcellular localization patterns. Of note, only the nucleus aggregation pattern of *MALAT1* showed striking colocalization with *miR145-5p*.

Finally, we performed multiple new experiments to address the following concerns and have included additional supporting data based on reviewer suggestions. All changes have been tracked with yellow highlights in the revised manuscript. We thank the reviewer again for the comments and suggestions, which significantly improved our study. Please see our point-by-point responses to reviewer comments in the following section.

My recommendation to the authors would be to look at other papers recently published on FISH signal amplification or multiplexed-omics technologies for guidance on a future manuscript (Multiplexed Detection of RNA using MERFISH and branch amplification- Xia et al., 2019). In my opinion, the manuscript suffers greatly from a general lack of quantification and necessary controls that would be required to support

the authors claims of enhanced specificity, sensitivity and efficiency of this FISH method. I found the manuscript would benefit from the inclusion of much more detail as this is a technical paper- a detailed methods section is lacking in this manuscript. A single π -probe for signal amplification is a desirable design in the method presented here, but there is insufficient data to currently support the authors claims as described in more detail below:

Response: Thank you for this comment. As suggested, we have carefully reread the related references and performed 18 additional experiments or analyses accordingly. With these additional data, we provide further supporting data with necessary controls, quantification analysis, and detailed information on the methods, as well as thoroughly address the reviewer's concerns in the revised manuscript (as shown in the following 18 figures).

To determine the specificity of π -FISH rainbow, we co-detected of mutually exclusive gene expression in mice brain slices (Fig. R3-3) and compared π -FISH and HCR (Fig. R3-4). We also confirmed the specificity of amplification probes (secondary and tertiary probes) (Fig. R3-2). Furthermore, we measured the off-target and false-positive rates (Fig. R3-3 and R3-10). The results obtained the high specificity of π -FISH rainbow. The scramble target probe for miRNA detection (Fig. R3-13) also supported the high π -FISH+ specificity.

For background-level issues, we included additional controls —unilateral target probes, no target probes, RNase treatment, and bacterial *dapB* probes— to measure the background noise resulting from π -FISH rainbow during the detection of *ACTB*, *PPIA*, *B2M* genes in HeLa cells, as well as *Ctgf*, *Penk*, and *Sst* genes in mouse brain tissues (Fig. R3-7). We used a series of gradient elution buffers to wash the samples after each hybridization step, minimizing noise and keep the positive signals. As shown in Fig. R3-7, there were negligible signal spots in the control cells and tissues. This high specificity and low background noise of π -FISH rainbow are most likely due to the stability of the π -FISH hybridization complex. To further verify the false positive rate of π -FISH rainbow for single or multiplexed detection, we introduced the off-target probe, nonspecific probes, and unilaterally specific probes as additional negative controls (Fig. R3-5). Rigorous quantification analyses were performed for all these experiments.

Moreover, consistent expression patterns detected using π -FISH rainbow and traditional hybridization of both high and low expression genes indicated high specificity, sensitivity, and efficiency of π -FISH rainbow. We also measured the

brightness, size, and variation in signal spots using different numbers of target probes based on the references (Fig. R3-1, R3-4, R3-6, and R3-8).

We have added the sequences of the amplification probes to Supplementary Table 1, the detailed schematic for π -FISH rainbow and π -FISH+ to Supplementary Fig. 1 and 13, and detailed descriptions of the methods in our revised manuscript.

The authors claim this method enhances signal intensity of individual FISH foci in varying modalities (1E,1G,3D-E, S1E). The authors measure signal intensity by taking a line average across the entire cell (S3, S7, S8). This does not in fact quantify signal intensity of individual foci, but rather measures the intensity across the specified area, which is simply brighter compared to controls because there are increased numbers of fish foci as shown by their results in Figure 1B. With this, the entire claim of enhanced signal intensity cannot be concluded.

Response: Thank you for the comment. We apologize for the unclear description.

For RNA detection, we measured signal intensity in two ways, as described below. On one hand, in Fig. 1e–g in the manuscript, we indeed measured the fluorescence intensity of all individual spots to compare the signal amplification ability of π -FISH rainbow with other single-molecule FISH methods. On the other hand, as shown in Supplementary Fig. 3 in the original manuscript (revised Supplementary Fig. 6), we also drew a series of lines across the cell and plotted the corresponding intensity profiles as the line passed. The results of the two measurement methods reveal that the intensity of the fluorescence signal generated by π -FISH rainbow was significantly higher than that generated using other methods.

For protein detection, we compare the signal intensity of π -FISH rainbow with traditional immunofluorescence using two measurements. First, since conventional immunostaining displays diffusive signals in the cell body (not dot-like signals), the fluorescent intensity of individual foci cannot be measured. Therefore, we measured the fluorescent intensity of single cells to compare signal amplification using π -FISH rainbow and conventional immunostaining (Fig. 3d–e in the manuscript). Second, for visual comparisons of fluorescence signal intensity and specificity between FISH and other methods, we measured the signal intensity profile by taking a line across the entire cell, and corresponding intensity profiles were plotted in Supplementary Fig. 7 and 8 in the original manuscript (revised Supplementary Fig. 10). Based on these data, we demonstrated high sensitivity and specificity of π -FISH strategy for protein detection.

Besides, as shown in Supplementary Fig. 1e in the original manuscript, *PPIA* mRNA was detected by π -FISH rainbow using different numbers of π target probe pairs

(2, 5, 10, 15, 20, and 25). Our data indeed indicated that the signal spot counts gradually increased as the number of target probes increased, and with 10–15 pairs of π target probes providing the peak number of signal spots (Fig. R3-1a and b, also Supplementary Fig. 2e and f in the revised manuscript). However, the increase in fluorescence intensity of individual signal spots was not due to the increase in the number of signal spots. Our data indicated that targeting a single signal spot with more probes can indeed increase the fluorescence intensity of individual signal spots (Fig. R3-1c, also Supplementary Fig. 2g in the revised manuscript).

Fig. R3-1. Analysis of mean spot counts and the fold-increase in brightness of individual spots with different numbers of π target probes.

(a) Detection of *PPIA* mRNA in HeLa cells with different numbers of probe pairs (2, 5, 10, 15, 20, and 25 pairs of π target probes). Scale bars, 10 μ m.

(b) Line chart of mean spot counts per cell from (a). n = 30 cells per group.

(c) The fold-increase in brightness of individual *PPIA* mRNA spots in HeLa cells from (a).

Figure 1B/S1D. What is the specificity of the primary π -probe binding to the target mRNA? What is the specificity of the tertiary and readout probes binding to secondary π -probes and to each other?

Response: To guarantee signal specificity and avoid background noise, we first performed sequence alignment of π target probes to minimize homology with other gene sequences. Second, the secondary amplification probes were designed to hybridize with the whole top region, from the left and right sides of bilateral π probes, to guarantee specificity. Thus, unilateral or nonspecific binding can be easily washed away under a series of stringent washing conditions. Our results in Supplementary Fig. R3-2 showed

that unilateral π target probe binding with secondary amplification probes did not generate any positive signals.

The hybridization sequences of all amplification probes (secondary and tertiary probes) and signal probes were derived from the MERFISH paper on readout sequences screened with BLAST+¹², which were aligned to ensure low homology and to minimize nonspecific binding. To further verify the specificity of these probes, we performed additional control experiments to detect the *ACTB* gene in HeLa cells in Fig. R3-2. *ACTB* mRNA signals were absent in all control groups with no target probes, secondary probes, and tertiary probes, suggesting a high specificity of all probes in π -FISH rainbow. We have added a figure showing this information as Supplementary Fig. 2j in the revised manuscript.

To further verify the specificity of π -FISH rainbow, we co-detected mutually exclusive gene expression (Fig. R3-3, also Supplementary Fig. 7e and f in the revised manuscript) and parallel comparison of π -FISH with HCR (Fig. R3-4) in mice brain slices. *Slc17a7* and *Gad1* are excitatory and inhibitory neuron markers, respectively⁸. *Vip* and *Sst* gene expression label mutually-exclusive subclasses of interneurons in the cerebral cortex⁹. As shown in Fig. R3-3, our results showed that these marker genes had distinct expression patterns. Higher-resolution images showed almost no *Gad1* expression (green) in *Slc17a7*⁺ neurons (red). Likewise, there was scarcely any *Slc17a7* expression in *Gad1*⁺ neurons. We observed similar mutually exclusive *Vip* and *Sst* expression patterns. In addition, the expression of *Ctgf* and *Gad1* genes detected using π -FISH is consistent with that detected using highly sensitive hybridization chain reaction (HCR)-FISH (Fig. R3-4). All these results demonstrate the high specificity and accuracy of π -FISH rainbow.

Fig. R3-2. Validation of the specificity of target probes, secondary probes, and tertiary probes of π -FISH rainbow.

Compared with the specific detection of the *ACTB* gene in HeLa cells, there was almost no noise spot in the control groups without target probes, secondary probes, and tertiary probes, respectively. Scale bars, 10 μ m.

Fig. R3-3. The specificity of π -FISH rainbow was verified by detecting the mutually exclusive expression of different neuron markers.

(a) Co-detection of *Sst* and *Vip*, two mutually exclusively expressed genes in different subclasses of interneurons in the mouse cerebral cortex, by π -FISH rainbow. The green and red arrows indicated *Sst*⁺ and *Vip*⁺ neurons, respectively. Scale bars, 10 μ m.

(b) Co-detection of inhibitory (*Gad1*) and excitatory (*Slc17a7*) neuron markers in mouse cerebral cortex by π -FISH rainbow. The green and red arrows indicated *Gad1*⁺ and *Slc17a7*⁺ neurons, respectively. Scale bars, 100 μ m.

Fig. R3-4. Simultaneous detection of *Ctgf* and *Gad1* by π -FISH rainbow and HCR.

(a-b) *Ctgf* (a) and *Gad1* (b) were simultaneously detected in the mouse brain tissues by π -FISH rainbow and HCR. White boxes represented the higher-resolution images. The red and green arrows indicated π -FISH rainbow and HCR signals, respectively. Scale bars, 100 μ m.

Figure S1D shows you can have non-specific, off target interactions between two unilateral π -fish probes with 0 complimentary bases. This suggests this method does not have high specificity. What are the off-target rates and thus, false positive rates of detection using this method? What about these rates when multiple RNA species are analyzed simultaneously during multiplexing later in the manuscript?

Response: Thank you for this comment. As with most detection methods, it is indeed extremely challenging to have zero background^{1, 13, 14, 15, 16}. To address these questions quantitatively, we designed three experiments to verify the off-target rates and false positive rates of π -FISH rainbow for single or multiplexed gene detection.

First, we performed single-molecule FISH (smFISH) and π -FISH rainbow to detect the long non-coding RNA XLOC_010514. Among the target probes used for both methods, we spiked one probe with an off-target probe binding to nucleus RNA^{17, 18} to test the noise signal level. Consistent with published results^{17, 18}, one off-target smFISH probe resulted in nonspecific signals in the nucleus, which can be avoided by excluding the off-target probe (Fig. R3-5a and b). Notably, for the π -FISH rainbow method, the off-target probe yielded almost no off-target signals in the nucleus (Fig.

R3-5c and d). This is probably because the design of the π target probe pairs increases the specificity and stability of the hybridization complex. Therefore, it can effectively eliminate false positive signals caused by the off-target probe.

Next, we used nonspecific probes to verify π -FISH rainbow off-target rates. We detected the *ACTB* gene and co-detected *ACTB*, *PPIA*, and *B2M* genes, accompanied by bacterial *dapB* probes for nonspecific binding control. As shown in Fig. R3-5e and f, the off-target rates, or false positive rates, for single and multiplex detection were 0.36% and 0.41%, respectively.

Finally, we introduced unilaterally specific probes to determine the off-target rate of π -FISH. *PPIA* mRNA in HeLa cells was detected using bilaterally specific and unilaterally specific π target probes, respectively (Fig. R3-5g). Likewise, we simultaneously detected *ACTB*, *PPIA*, and *B2M* mRNA in HeLa cells using bilaterally specific and unilaterally specific π target probes (Fig. R3-5h). As shown in Fig. R3-5g and h, the off-target rates of single and multiplex detection using unilaterally specific binding were 0.48% and 0.51%, respectively. Overall, the off-target rate, or false positive rate, of π -FISH rainbow was less than 0.51%. We have added this information as Supplementary Fig. 5 in the revised manuscript.

Fig. R3-5. Verification of the off-target and false positive rates in single and multiplexed detection of π -FISH rainbow.

(a-b) Detection of XLOC_010514 by unamplified FISH. One off-target probe ('rogue' probe, spiked in red) caused false positive signals in the nucleus as indicated by the red arrows (a). Removal of the 'rogue' probe eliminated the false positive signals in the nucleus (b). Scale bars, 5 μ m.

(c-d) Detection of XLOC_010514 by π -FISH rainbow. No false positive signals were detected in the nucleus despite the 'rogue' probe design when full π target probes (c) or half π target probes (d) were added. Scale bars, 5 μ m.

(e-f) False positive rates of single (e) and multiplexed (f) detection of π -FISH rainbow were verified by detecting *ACTB* expression alone and simultaneously detecting *ACTB*, *PPIA*, and *B2M* expression, respectively along with nonspecific probes (bacterial *dapB* probes) addition. The false positive rate of single detection of π -FISH rainbow is the ratio of the total false positive spot counts of *dapB* (green) over the total spot counts of *ACTB*. The false positive rate of multiplexed detection of π -FISH rainbow is the ratio of the false positive spot counts (total spot counts of *ACTB* (green), *PPIA* (cyan), *B2M* (red), and *dapB* (encoded by three channels) minus the total spot counts of *ACTB*, *PPIA*, and *B2M*) over the total spot counts of *ACTB*, *PPIA*, and *B2M*. Scale bars, 10 μ m.

(g-h) The false positive rate of single (g) and multiplexed detection (h) of π -FISH rainbow were further verified by detecting the *PPIA* gene expression alone and simultaneously detecting *ACTB*, *PPIA*, and *B2M* genes expression, with bilaterally specific and unilaterally specific π target probes. The false positive rate for single detection of *PPIA* by π -FISH rainbow is the total spot counts of *PPIA* with unilaterally specific π target probes over the total spot counts of *PPIA* with bilaterally specific π target probes. The false positive rate of multiplexed detection of π -FISH rainbow is the total spot counts of *ACTB* (red), *PPIA* (cyan), and *B2M* (cyan) with unilaterally specific π target probes over the total spot counts of *ACTB*, *PPIA*, and *B2M* with bilaterally specific π target probes. Scale bars, 10 μ m. White numbers in the images of (e) to (h) indicated false positive rates.

More generally, what is the signal intensity variability and detection efficiency of the π -probes methods?

Response: The secondary (S) and tertiary (T) amplification probes of π -FISH rainbow contain 16 and 8 repeat sequences, respectively. Thus, the fluorescence signal would be theoretically amplified 16×8 -fold. Increasing degrees of amplification would likely result in larger spot size and more variability in spot brightness, which may subsequently impact the ability to distinguish and identification spots. Thus, the ideal amplification strategy would set the best balance between amplification, spot size, and brightness variability. To evaluate whether the 16×8 amplification strategy affects the ability to distinguish and identify spots using π -FISH rainbow, we conducted and compared the amplification performance of unamplified, 16×8 amplified, 16×4 amplified, 8×8 amplified, and 8×4 amplified samples using 15 π -target probes (Fig. R3-6a). As shown in Fig. R3-6b-d, we measured the fold-increase in spot brightness, spot sizes (full width at half maximum, FWHM) of individual spots, and the variation in spot brightness in each group. The results indicated that 16×8 amplification had maximum signal brightness (Fig. R3-6b). Notably, the spot sizes were comparable (~ 370 nm) across unamplified, 16×8 amplification, 16×4 amplification, 8×8 amplification, and 8×4 amplification groups (Fig. R3-6c). The coefficient of variation in spot brightness was similar for all amplifications, indicating that the 16×8 amplification does not increase the variation in spot brightness (Fig. R3-6d). Thus, we chose 16×8 amplification to detect the π -FISH rainbow signal.

Next, we evaluated the detection efficiency of π -FISH rainbow from three aspects. First, we compared π -FISH rainbow with HCR and smFISH by detecting high-low abundance transcripts (*ACTB*, *PPIA*, *B2M*, and *MTOR* genes). The fluorescence signal intensity of individual spots and the number of signal spot counts generated using π -FISH rainbow was significantly higher than that generated using the other methods (Fig.

R3-7).

To further verify the efficiency of π -FISH rainbow in tissue samples, we attempted to detect high and low-abundance transcripts based on ISH results from the Allen Institute of Brain Science⁷ (Fig. R3-8). π -FISH rainbow faithfully reproduced the spatial distribution of both high (*Cux2* and *Pcp4*) and low (*Lrmp* and *Ptpru*) abundance genes, indicating the sensitivity and reliability of π -FISH rainbow in analysing tissue samples.

Finally, to assess the noise level associated with π -FISH rainbow, we measured *ACTB*, *PPIA*, and *B2M* genes in HeLa cells (Fig. R3-9a and b) and *Ctgf*, *Penk*, and *Sst* genes in mouse brain tissues (Fig. R3-9c and d) using different negative controls, including unilateral target probes, no target probes, RNase treatment, and bacterial *dapB* probes. As shown in Fig. R3-9e-h, there was almost no signal spot in the negative control groups. To verify the off-target rate of π -FISH rainbow for single and multiplexed gene detection, we introduced the off-target probe, nonspecific probes (bacterial *dapB* probes), and unilaterally specific probes, respectively. The off-target rate of π -FISH rainbow was less than 0.51% (Fig. R3-5). Together, these data validate the high detection efficiency and sensitivity of the π -FISH rainbow method.

Fig. R3-6. π -FISH rainbow dramatically increases signal brightness without increasing spot size.

(a) Detection of *PPIA* mRNA using 15 π target probes with un-amplification, 16 × 8 amplification, 16 × 4 amplification, 8 × 8 amplification, and 8 × 4 amplification, respectively. Scale bars, 10 μ m.

(b) The spot sizes were measured for un-amplification, 4 × 4 amplification, 8 × 8 amplification,

16 × 4 amplification, and 16 × 8 amplification groups from (a). The width (full width at half maximum) was determined by Gaussian fitting of RNA spots.

(c) The fold-increase in average brightness of individual *PPIA* mRNA spots in 4 × 4 amplification, 8 × 8 amplification, 16 × 4 amplification, and 16 × 8 amplification samples relative to the brightness observed for un-amplification samples.

(d) The fold-increase in brightness of individual *PPIA* mRNA spots in HeLa cells with different probe pairs (2, 5, 10, 15, 20, and 25 pairs of π target probes).

Fig. R3-7. Comparison of hybridization efficiency of π -FISH rainbow with HCR, smFISH, and smFISH-FL.

(a-c) Comparisons of the hybridization efficiency among π -FISH rainbow, HCR, smFISH, and smFISH-FL (probes covered the full-length transcript) by detecting *ACTB* mRNA in HeLa cells (a). Scale bars, 10 μ m. 16 target probes (for π -FISH rainbow, HCR, and smFISH) and 23 target probes (for smFISH-FL) were used. The signal spots per cell and fluorescent signal intensity per signal spot of π -FISH rainbow, HCR, smFISH, and smFISH-FL were counted and are illustrated in (b) and (c), respectively. Data were expressed as mean \pm s.e.m. Two-tailed unpaired Student's *t* test was used to compare the difference between the two groups. ** $P < 0.01$; **** $P < 0.0001$; ns, not significant. (b)

π -FISH rainbow, n = 47 cells; HCR, n = 53 cells; smFISH-FL, n = 33 cells; smFISH, n = 40 cells. smFISH-FL vs. smFISH, $P = 0.8616$; HCR vs. smFISH-FL, $P = 1.26 \times 10^{-5}$; π -FISH rainbow vs. HCR, $P = 0.0081$; π -FISH rainbow vs. smFISH-FL, $P = 1.04 \times 10^{-9}$. (c) π -FISH rainbow, n = 8362 spots; HCR, n = 9564 spots; smFISH-FL, n = 14234 spots; smFISH, n = 14241 spots.

(d) The signal intensity of *ACTB* mRNA was detected by π -FISH rainbow, HCR, smFISH-FL, and smFISH in HeLa cells, respectively. Equal concentrations of target probes with equal lengths were used for all methods except smFISH-FL, where the probes targeted the whole transcript. As illustrated, the straight lines were drawn across the cells, of which the intensity profiles were plotted (*ACTB* mRNA, red curve lines; DAPI, blue curve lines). Scale bars, 10 μ m.

(e) Detection of *PPIA*, *B2M*, and *MTOR* mRNA by π -FISH rainbow, HCR, and smFISH, respectively. Scale bars, 10 μ m.

Fig. R3-8. Validation of highly and lowly expressed genes by π -FISH rainbow.

(a-d) The gene expression pattern of π -FISH rainbow was consistent with the results of the Allen

Institute for Brain Science (AIBS), both for the detection of highly expressed genes *Cux2* (a) and *Pcp4* (b) and lowly expressed genes *Lrmp* (c) and *Ptpru* (d) in mouse brain tissues. Scale bars, 50 μm (a and b); 100 μm (c and d).

Fig. R3-9. Background of π -FISH rainbow in cells and tissues.

(a-b) The background of π -FISH rainbow in cells was verified by *in situ* detection of *ACTB* (a) and co-detection of *ACTB*, *PPIA*, and *B2M* (b) along with multiple negative controls, including unilateral target probes, no target probes, bilateral target probes with RNase treatment, and bacterial *dapB* probes in HeLa cells, respectively. Scale bars, 10 μ m.

(c-d) The background of π -FISH rainbow in tissues was verified by *in situ* detection of *Ctgf* (c) and co-detection of *Ctgf*, *Penk*, and *Sst* (d) along with multiple negative controls, including unilateral target probes, no target probe, bilateral target probes with RNase treatment, and bacterial *dapB* probes in mouse brain sections, respectively. (i-v) Higher magnification of square regions in the upper panels of (c) and (d) were shown in the lower panels. Scale bars, 100 μ m (top) and 20 μ m (bottom).

(e-h) The histogram of spot counts per cell for positive signals and background noise from (a-d). n = 30 cells per group.

Figure 1D/S1E. Does increasing the number of π -FISH for a given mRNA alter the FISH foci size? Moreover, what is the size of a single FISH foci using only one π -probe? Increased foci size can result in the cytoplasmic crowding of RNA FISH signal preventing proper identification/segmentation of individual foci during analysis i.e Fig1E- π -probe condition. This is undesirable as this will impair proper mRNA copy number counts- see Xia et al., 2019 branch DNA amplification paper.

Response: Thank you for this important comment. To address whether increasing the number of π -FISH for a given mRNA alters the FISH foci size, we measured the size of individual spots generated using 1, 2, 5, 10, 15, 20, and 25 π target probes (Fig. R3-10a and b). As shown in Fig. R3-10c, the size of single fluorescent spots generated using 1, 2, 5, 10, 15, 20, and 25 π target probes were comparable, at approximately 370 nm. This data is consistent with the previous study, in which Xia et al. provided an excellent combination of the MERFISH and branched DNA (bDNA) amplification approaches, producing a significant increase in the signal without increasing the size of individual spots¹⁹.

Fig. R3-10. The fluorescent spot sizes of π -FISH rainbow using different numbers of π target probes.

(a-b) Detection of *PPIA* mRNA in HeLa cells with 1, 2, 5, 10, 15, 20, and 25 π target probes, respectively. Scale bars, 10 μ m.

(c) The spot sizes were measured in the hybridization with different numbers of probe pairs (1, 2, 5, 10, 15, 20, and 25 π target probes). The width (full width at half maximum) was determined by Gaussian fitting of RNA spots.

Figure 1E. smFISH-FL is reported to give the most accurate mRNA counts compared to RNA-sequencing methods. I am therefore somewhat surprised and confused that both HCR and π -FISH yield almost double the RNA counts per cell of smFISH-FL method? Can the authors compare their counts to what others have found? As well, this may indicate off-target effects.

Response: We agree with the reviewer that smFISH-FL is reported as the most accurate mRNA count method. However, the absence of signal amplification and the potentially incomplete hybridization of all probes due to hybridization probability can result in insufficient signal detection. Although highly sensitive imaging techniques, such as TIRF, can capture weak signals, the imaging area and Z-axis depth are unsuitable for most spatial omics studies. Thus, not all mRNA can be efficiently detected using smFISH-FL. In this scenario, amplification can significantly enhance the intensity of weak signals in smFISH. As described by Xia et al., 2019, increasing branch amplification can improve detection efficiency. In this study, the detection efficiency of π -FISH rainbow was significantly higher than that of the smFISH-FL when we used the same number of target probes due to the strong amplification capability (Fig. 1f in the

manuscript). Moreover, as our data demonstrated, there were negligible off-target effects in π -FISH rainbow.

Figure 2-3

Can the authors explain what channel arithmetic entails as a decoding scheme and present its misidentification rates and detection efficiency? Is there an error-robust correction strategy the authors could use?

Response: Our decoding strategy is based on combining different fluorescence channels. As described in the manuscript, “In theory, the combination of four fluorescence signal probes can generate 15 ($C_4^1+C_4^2+C_4^3+C_4^4$) different signal codes and differentiate 15 genes in one round” (lines 144 to 146). Our data demonstrated that the ratio of overlapping two and three fluorescence signals during the signal probe combination experiment was 99.14% and 99.01% (Fig. R3-11a and b), respectively, suggesting that combining different signal probes for multiplex target detection could be highly reliable.

To further evaluate the efficiency of multiple detections, we compared multi-channel detections of *Gad1* and *Vip* genes in mouse brain tissues with single-channel detections (Fig. R3-11c-e). As shown in Fig. R3-11f and g, the results showed that multi-channel *Gad1* and *Vip* gene detection efficiencies were 99.03% and 99.10% of single-channel detection, respectively, indicating high efficiency and accuracy of π -FISH rainbow in multiplex detection and decoding.

MERFISH is a breakthrough single-molecule imaging approach that simultaneously locates thousands of RNA species using dozens of iterative labeling and imaging performed using error-robust encoding schemes¹². In this study, π -FISH rainbow was designed for low to medium throughput. Theoretically, up to 15 genes can be detected simultaneously in one round by combining channels. The number of targets scale linearly with the number of hybridization rounds ($15 \times N$, N means hybridization rounds). Due to the low-medium throughput of π -FISH rainbow in each round, there are relatively fewer signal spots in individual cells, making signal analysis much less challenging. Since barcodes are not used, images from each round can be analysed separately, avoiding the difficulty of single fluorescent signal spot alignment and correction between multiple rounds. Therefore, we used the Imaris software for signal decoding and did not use an error-robust strategy, considering the low-medium throughput, high efficiency, and accuracy of π -FISH rainbow during multiplex detection and decoding. However, if π -FISH rainbow is used for high throughput detection by multiplexing sequential hybridization, it will need crowded signal

alignment and more error-robust correction. We have added this information as Supplementary Fig. 3 in the revised manuscript.

Fig. R3-11. Efficiency and accuracy of π -FISH rainbow in multiplexed detection and decoding.

(a-b) *In situ* detection of *ACTB* mRNA with two (b) or three (c) fluorescence signal probes. The overlap ratios were 99.14% (b) and 99.06% (c), respectively. n = 30 cells. Ch 1, channel 1. Ch 2, channel 2; Ch 3, channel 3. Scale bars, 10 μ m.

(c-d) *In situ* detection of *Gad1* (d) and *Vip* (e) genes expression with triple channels in mouse brain tissues. Scale bars, 100 μ m (lower magnification) and 10 μ m (high magnification).

(e) *Gad1* mRNA (left) and *Vip* mRNA (right) were detected with a single channel in mouse brain tissues, respectively. Scale bars, 100 μ m (lower magnification) and 10 μ m (high magnification).

(f-g) The efficiency and accuracy of *Gad1* (f) and *Vip* (g) in multiplexed detection were 99.03% and 99.10%, respectively. n = 30 cells.

Refer to above about issues surrounding the quantitation method for signal intensity.

Response: Because traditional immunofluorescence signals do not display as single spots, we compared fluorescence signal intensities of whole cells generated using traditional immunofluorescence and π -FISH rainbow in pol II protein detection (Fig. 3d and e in the manuscript). Stronger protein fluorescence signals were observed with π -FISH rainbow. We have added this information as Supplementary Fig. 10 in the revised manuscript.

How many π -probes are used per ssDNA oligo conjugated antibody? What are the off-target, false-positive and misidentification rates for all the oligos presented during this design?

Response: Each antibody was conjugated to a 47-nt ssDNA oligonucleotide and then hybridized with a 166-nt oligonucleotide extension probe for four π -probes binding. We verified off-target and nonspecific binding of π -FISH rainbow in protein detections using multiple negative controls, including no specific antibody, no ssDNA, no extension probe, non-target antibody (antibody against classical swine fever virus E2), scramble ssDNA, and scramble extension probe (Fig. R3-12a–h). Of note, the E2 protein is not expressed in HeLa cells. As shown in Fig. R3-12b–h, there was almost no noise signal in the control groups. We further validated the false-positive rates by co-detecting Pol II and E2 proteins in HeLa cells (Fig. R3-12i). The false positive rate was 0.25%. We have added this figure as Supplementary Fig. 11 in the revised manuscript.

Fig. R3-12. The specificity and efficiency of π -FISH rainbow in protein detection.

(a) Detection of pol II protein in HeLa cells by π -FISH rainbow using ssDNA conjugated pol II antibody and corresponding extension probe. Scale bar, 10 μ m.

(b-d) Detection of pol II protein in HeLa cells by π -FISH rainbow without pol II antibody, specific ssDNA, and specific extension probe, respectively. Scale bars, 10 μ m.

(e) Detection of pol II protein in HeLa cells by π -FISH rainbow without antibody, ssDNA, and extension probe. Scale bar, 10 μ m.

(f-h) Detection of pol II protein in HeLa cells (without E2 expression) by π -FISH rainbow using ssDNA conjugated pol II antibody and corresponding extension probe replaced by E2 antibody, scramble ssDNA, and scramble extension probe, respectively. Scale bars, 10 μ m.

(i) The off-target and false-positive rate was verified by co-detection of pol II protein (red) and E2 protein (without E2 expression, green) by π -FISH rainbow in HeLa cells. The percentage of false-

positive rate of π -FISH rainbow in protein detection was 0.25% from (i). n = 30 cells. Scale bar, 10 μ m.

Figure 4

S9. To determine the specificity of the DNA π -probes to centromeres, the authors need to co-localize the π -probe signal with a centromere marker.

Response: As suggested, we performed a co-localization assay of π -FISH rainbow and the centromere marker CENPA to determine the specificity of the DNA π -probes binding to centromeres. As shown in Fig. R3-13, the π -FISH rainbow signal (green) overlapped with that of the centromere marker (red).

Fig. R3-13. Co-localization of centromere signals detected by π -FISH rainbow and the centromere antibody CENPA.

(a) Diagram of centromere in chromosome 2 (left), and co-localization of centromere signals (right) detected by π -FISH rainbow (green) and the centromere antibody CENPA (red). Scale bar, 2.5 μ m.

Figure 5

In the opening sentence the authors claim 1KB is required for current FISH methods which is not accurate. 500bp for mRNA has been achieved; see Xia et al., 2019- Spatial transcriptome profiling by MERFISH

Response: Thank you for the correction. Xia et al. used an overlapping encoding-probe design to detect RNAs as short as 500 bp in length using 48 encoding probes per RNA. We have changed our description from “1 kb” to “500 bp” in the revised manuscript and cited the study by Xia et al. (2019) for reference²⁰.

Similar to concerns presented in Figure 1, the introduction of HCR here requires the presentation of appropriate controls. How was HCR tunability achieved, number of repetitive elements following HCR that were used in final probe design, off target rates and specificity of readout probes to HCR repetitive elements, information about the self-folding hairpin sequence, signal intensity fold changes associated with HCR, signal intensity variation, detection efficiency, foci size etc

Response: For π -FISH+, the π -FISH rainbow signal probe is replaced with the split-initiator probes and self-folding hairpins (hairpin H1 and H2) to amplify the signal further. According to HCR 3.0, a pair of split-initiator probes form HCR initiator I1, which hybridize to the input domain of hairpin H1 and subsequently trigger chain reactions of hairpin H1 and H2¹. The sequences of split-initiator probes and hairpin probes can be found in Supplementary Table 1. In the 16×8 amplification scheme, each tertiary amplification probe hybridizes with four pairs of split-initiator probes. Thus, the π -FISH+ strategy using one pair of π target probes can theoretically initiate 64 HCR reactions, triggering orthogonal HCR amplification cascades that increase amplification polymers over time. To verify signal intensity variation and spot size, we performed HCR amplification using one pair of π target probes for 1, 2, 4, 8, and 16 h to generate varying amplification polymers (Fig. R3-14a). As shown in Fig. R3-14b and c, we measured the fluorescence intensity and size (FWHM) of individual spots. Our data showed that HCR amplification for 16 h resulted in maximal signal brightness without increasing spot size, making it appropriate for π -FISH+ detection.

To determine the variation in spot brightness in each group, we measured the coefficient of variation with HCR amplification for 1, 2, 4, 8, and 16 h (Fig. R3-14d). Notably, we found that the coefficient of variation was similar for all amplifications, indicating that HCR amplification does not increase the variation in spot brightness.

Next, we verified the off-target rate and specificity of the HCR reactions using several negative controls, including no initiator probes, scramble probes, and hairpin H1 or H2 (Fig. R3-14e). As shown in Fig. R3-14f and g, we co-detected *PPIA* and *dapB* (bacterial gene) mRNAs in HeLa cells using π -FISH+ and calculated the off-target rate, which was 0.36%.

Finally, we attempted to determine the detection efficiency of π -FISH+ by comparing it with smFISH-FL. However, this experiment did not work because π -FISH+ uses only one pair of π target probes, whereas smFISH-FL requires multiple probes for visualization. Thus, as shown in Fig. R3-14h, we compared the detection efficiency of π -FISH+ with that of π -FISH using one pair of π target probes, as the detection efficiency of π -FISH has been determined through extensive comparison. Our data showed that the detection efficiency of π -FISH+ was similar to that of π -FISH (Fig. R3-14i). However, the intensity of the spot was stronger than that of the π -FISH (Fig. R3-14j). We have added this information as Supplementary Fig. 14 in the revised manuscript.

Fig. R3-14. Validation of efficiency, specificity, and tunability of HCR in π -FISH+ strategy.

(a) Detection of *PPIA* mRNA in HeLa cells with one pair of π target probes by π -FISH+, in which the HCR amplification was performed for 1 h, 2 h, 4 h, 8 h, and 16 h, respectively. Scale bars, 10 μ m.

(b) The fluorescence intensity of individual spots from (a).

(c) The spot sizes were measured in different HCR amplification schedules from (a). The width (FWHM) was determined by the Gaussian fitting of RNA spots.

(d) The variation coefficients of spot brightness when HCR amplifies for 1 h, 2 h, 4 h, 8 h, and 16 h, respectively. Error bars in (b–d) represent the standard deviation across three replicates.

(e) Detection of *PPIA* mRNA in HeLa cells by π -FISH+ with specific HCR split-probe, without HCR split-probe, with HCR split-probe containing scramble initiator, and with hairpin H1 or H2 only, respectively. Scale bars, 10 μ m.

(f-g) The false positive rate of HCR amplification in π -FISH+ was verified by co-detection of *PPIA* (red) and *dapB* (nonspecific binding, green) in HeLa cells (f), and the false positive rate was 0.36% (g). Scale bars, 10 μ m.

(h) Detection of *ACTB* mRNA in HeLa cells with one pair of π target probes by π -FISH rainbow (left) and π -FISH+ (right), respectively. Scale bars, 10 μ m.

(i) The spot counts per cell of *ACTB* mRNA detected by π -FISH+ with one pair of π target probes were comparable to that by π -FISH.

(j) The fluorescence intensity of *ACTB* mRNA detected by π -FISH+ with one pair of π target probes was significantly higher than that of π -FISH. $P < 1 \times 10^{-15}$. ****P < 0.0001.

The above controls also apply to the 54bp extension probe for miR-145.

Response: As suggested, we performed several controls, including a scramble extension probe and no extension probe, to further verify the specificity of the 54 bp extension probe. As shown in Fig. R3-15a–c, there is almost no background noise, confirming the specificity of the extension probe in detecting miRNA. While high levels of *miR145-5p* signals were detected in the experimental group, the signals were barely observed in the control groups, indicating that π -FISH+ can detect miRNA effectively and accurately. Moreover, the signal intensity, spot size, and detection efficiency of π -FISH+ were validated. We have added this figure as Supplementary Fig. 15 in the revised manuscript.

Fig. R3-15. Validation of the specificity and efficiency of π -FISH+ in miRNA detection.

(a) The *miR145-5p* was detected using a specific extension probe as a positive control. Scale bar, 2.5 μ m.

(b-c) The specificity of the miRNA complementary sequence within the extension probe was verified with a scramble extension probe (nonspecific binding, b) and no extension probe (c) for the hybridization. Scale bars, 2.5 μm .

Figure 5B Malat1 staining looks dramatically different than controls shown in S10ab. Thus, this cannot be used to support the efficiency and accuracy of this method.

Response: Thank you for the comment. As we mentioned in the original manuscript, our method revealed three main patterns of lncRNA *MALATI* distribution: mainly in the nucleus, mainly in the cytoplasm, and evenly distributed in both the nucleus and cytoplasm. If we investigate the distribution of *MALATI* in the nucleus further, we find two patterns: some cells show a more even distribution in the nucleus, while others show larger aggregated *MALATI* signals. Interestingly, aggregated *MALATI* showed more striking colocalization with *miR145-5p*. Thus, we only showed the colocalization of *miR145-5p* with the aggregated *MALATI* signals in Fig. 5b in the original manuscript (Fig. R3-16). It would be of great interest to further investigate the dynamic transition between different subcellular distributions of *MALATI* and the associated functions, especially in conjunction with *miR145-5p*. These data support the high efficiency of π -FISH+ for simultaneous detection of lncRNA and miRNA. We have added this figure as Supplementary Fig. 16c in the revised manuscript.

Fig. R3-16. Co-detection of *miR-145* and *MALATI* by π -FISH+.

The *miR145-5p* and its sponge lncRNA *MALATI* were co-detected by π -FISH+. Only the distribution of *miR145-5p* under the nuclear expression pattern of *MALATI* was shown here. Scale bars, 2.5 μm .

Figure 5G; needs controls outlined above

Response: As suggested, we have added more negative controls in this experiment. To further verify the ability of π -FISH+ to detect the *ARV7* splice variant, we collected blood from two additional patients with prostate cancer and analysed *ARV7* in 15 and 18 circulating tumour cells (CTCs) from these patients. As shown in Fig. R3-17, the *ARV7* splice variant was detected in CTCs but not in healthy human blood cells, validating the integrity of this strategy for diagnosis of anti-androgen therapy resistance. We have added this Fig. R1-17c-e as Supplementary Fig. 17 in the revised manuscript.

Fig. R3-17. Detection of *ARV7* splice variant in circulating tumor cells from prostate cancer patients.

(a) Schematic of CTCs isolated from prostate cancer patients and detected by π -FISH+.

(b) Diagram of probe design for identifying the *ARV7* variant by simultaneously labeling locus 1 (in *ARV1*, *ARV2*, *ARV4*, and *ARV7*) and locus 2 (in *ARV5* and *ARV7*). The *K19* gene was used as an internal control.

(c) Detection and decoding of *K19*, *ARV7* locus 1, and *ARV7* locus 2. The co-localization of the locus 1 and 2 signals was identified as *ARV7* (right). Scale bars, 5 μm .

(d) Application of π -FISH+ for *ARV7* detection in multiple CTCs from two more prostate cancer patients. Scale bars, 5 μm .

(e) Healthy human blood cells were used as negative controls. Scale bars, 5 μm .

For the CRISPR experiments, more evidence/descriptions of these mutations and methodology needs to be presented. Moreover, this method only works if mutations are very large and not small as the authors have suggested given they are investigated a 2KB deletion.

Response: CRISPR technology can knock out large fragments of more than 10 kb²¹, small fragments of less than 2 kb²², and even single bases in the genome sequence²³. To confirm the knocked out of the sequence (1,371 bp), we performed agarose gel electrophoresis and Sanger sequencing. Fig. R3-18a and b unambiguously showed that we successfully knocked out the 1,371 bp sequence of *LRRK2* in A549 cells. We have added a more detailed description of gene editing via CRISPR technology in the revised manuscript (lines 786 to 800).

As π -FISH+ can significantly amplify the signal after target probe hybridization by sequential amplification probe and HCR, it can be used to detect small RNA and DNA sequences. To validate that π -FISH+ can indeed detect small genomic loci, we detected the genomic breakpoint, which is independent of the length of the knock-out sequence. As shown in Fig. 5h and i in the manuscript, we designed two π target probes, one binding to the genomic breakpoint and the other targeting the normal sequence upstream of the breakpoint, as a reference point, to confirm the ability of π -FISH+ to detect short fragments. Using this strategy, our data showed that we could detect this small deletion.

Fig. R3-18. Identification of the sequence deletion in A549 cells knocked out by CRISPR.

(a) Agarose gel electrophoresis of the PCR products indicated the correct fragment size for deletion in A549-KO cells. M: DL2000 DNA Marker. WT: 1895 bp. KO: 524 bp.

(b) Schematic and Sanger sequencing results of A549 wild-type (A549-WT, up) cells and A549 knock-out (A549-KO, bottom) cells confirmed the deleting sequence

References:

1. Choi HMT, Schwarzkopf M, Fornace ME, Acharya A, Artavanis G, Stegmaier J, *et al.* Third-generation in situ hybridization chain reaction: multiplexed, quantitative, sensitive, versatile, robust. *Development* 2018, **145**(12).
2. Codeluppi S, Borm LE, Zeisel A, La Manno G, van Lunteren JA, Svensson CI, *et al.* Spatial organization of the somatosensory cortex revealed by osmFISH. *Nature methods* 2018, **15**(11): 932-935.
3. Fang R, Xia C, Close JL, Zhang M, He J, Huang Z, *et al.* Conservation and divergence of cortical cell organization in human and mouse revealed by MERFISH. *Science* 2022, **377**(6601): 56-62.
4. Takei Y, Yun J, Zheng S, Ollikainen N, Pierson N, White J, *et al.* Integrated spatial genomics reveals global architecture of single nuclei. *Nature* 2021, **590**(7845): 344-350.
5. Bintu B, Mateo LJ, Su JH, Sinnott-Armstrong NA, Parker M, Kinrot S, *et al.* Super-resolution chromatin tracing reveals domains and cooperative interactions in single cells. *Science* 2018, **362**(6413).
6. Landry JJ, Pyl PT, Rausch T, Zichner T, Tekkedil MM, Stütz AM, *et al.* The genomic and transcriptomic landscape of a HeLa cell line. *G3* 2013, **3**(8): 1213-1224.
7. Lein ES, Hawrylycz MJ, Ao N, Ayres M, Bensinger A, Bernard A, *et al.* Genome-wide atlas of gene expression in the adult mouse brain. *Nature* 2007, **445**(7124): 168-176.
8. Wang Y, Eddison M, Fleishman G, Weigert M, Xu S, Wang T, *et al.* EASI-FISH for thick tissue defines lateral hypothalamus spatio-molecular organization. *Cell* 2021, **184**(26): 6361-6377 e6324.
9. Gouwens NW, Sorensen SA, Berg J, Lee C, Jarsky T, Ting J, *et al.* Classification of electrophysiological and morphological neuron types in the mouse visual cortex. *Nature neuroscience* 2019, **22**(7): 1182-1195.
10. Zhang M, Eichhorn SW, Zingg B, Yao Z, Cotter K, Zeng H, *et al.* Spatially resolved cell atlas of the mouse primary motor cortex by MERFISH. *Nature* 2021, **598**(7879): 137-143.
11. Tripathi V, Ellis JD, Shen Z, Song DY, Pan Q, Watt AT, *et al.* The nuclear-retained noncoding RNA MALAT1 regulates alternative splicing by modulating SR splicing factor phosphorylation. *Molecular cell* 2010, **39**(6): 925-938.

12. Chen KH, Boettiger AN, Moffitt JR, Wang S, Zhuang X. RNA imaging. Spatially resolved, highly multiplexed RNA profiling in single cells. *Science* 2015, **348**(6233): aaa6090.
13. Goh JYL, Chou N, Seow WY, Ha N, Cheng CPP, Chang YC, *et al.* Highly specific multiplexed RNA imaging in tissues with split-FISH. *Nature methods* 2020, **17**(7): 689–693.
14. Shah S, Lubeck E, Schwarzkopf M, He TF, Greenbaum A, Sohn CH, *et al.* Single-molecule RNA detection at depth by hybridization chain reaction and tissue hydrogel embedding and clearing. *Development* 2016, **143**(15): 2862–2867.
15. Cao D, Wu S, Xi C, Li D, Zhu K, Zhang Z, *et al.* Preparation of long single-strand DNA concatemers for high-level fluorescence in situ hybridization. *Communications biology* 2021, **4**(1): 1224.
16. Choi HM, Beck VA, Pierce NA. Next-generation in situ hybridization chain reaction: higher gain, lower cost, greater durability. *ACS nano* 2014, **8**(5): 4284–4294.
17. Deng R, Zhang K, Sun Y, Ren X, Li J. Highly specific imaging of mRNA in single cells by target RNA-initiated rolling circle amplification. *Chemical science* 2017, **8**(5): 3668–3675.
18. Cabili MN, Dunagin MC, McClanahan PD, Biaesch A, Padovan-Merhar O, Regev A, *et al.* Localization and abundance analysis of human lncRNAs at single-cell and single-molecule resolution. *Genome biology* 2015, **16**(1): 20.
19. Xia C, Babcock HP, Moffitt JR, Zhuang X. Multiplexed detection of RNA using MERFISH and branched DNA amplification. *Scientific reports* 2019, **9**(1): 7721.
20. Xia C, Fan J, Emanuel G, Hao J, Zhuang X. Spatial transcriptome profiling by MERFISH reveals subcellular RNA compartmentalization and cell cycle-dependent gene expression. *Proceedings of the National Academy of Sciences of the United States of America* 2019, **116**(39): 19490–19499.
21. Hao H, Wang X, Jia H, Yu M, Zhang X, Tang H, *et al.* Large fragment deletion using a CRISPR/Cas9 system in *Saccharomyces cerevisiae*. *Analytical biochemistry* 2016, **509**: 118–123.
22. Kanca O, Zirin J, Garcia-Marques J, Knight SM, Yang-Zhou D, Amador G, *et al.* An efficient CRISPR-based strategy to insert small and large fragments of DNA using short homology arms. *eLife* 2019, **8**.

23. Eid A, Alshareef S, Mahfouz MM. CRISPR base editors: genome editing without double-stranded breaks. *The Biochemical journal* 2018, **475**(11): 1955-1964.

Reviewers' Comments:

Reviewer #1:

Remarks to the Author:

The authors have added a substantial amount of control data and addressed the reviewers' comments in the revised manuscript. The addition of schematics and clarifications in the methods should help other laboratories implement the n-FISH and n-FISH+ methods. The description of n-FISH/n-FISH+ as new in situ detection methods should be valuable to laboratories that may not have the ability to use other higher throughput in situ analysis methods. The increased sensitivity should be useful for detection of rare or small RNAs, short exons, and sequence variants.

Specific comments:

Fig. S2. FWHM is not defined in legend (full width at half maximum?).

Fig. S3g, h: why is multichannel detection more efficient than single channel for both probes? If multichannel spots must be detected with all colors to be scored, they should be less frequent than the single channel detection frequencies. If multichannel spots are being scored as positive with missing channels (e.g., 2 of 3 channels), that could lead to a higher percentage of detection, but it would be misleading since a multichannel spot with one color missing could lead to incorrect probe identification when multiple targets are detected together.

Fig. S16c. The additional examples of miR-145-5p detection, as well as the additional controls are helpful. The correlation between miR-145-5p and Malat1 subcellular localization is not as apparent as in Fig. 5b (other than that both are in the nucleus), but the experiment still demonstrates the ability to detect overlapping expression between miRNAs and lncRNAs in cells.

Fig. S17 may be easier for readers to understand if the figure legend states the conclusions from the figure: why is detection of the two ARV7 loci of interest (currently only stated in the main text)?

Reviewer #2:

Remarks to the Author:

We thank the authors for their detailed revisions which go a long way towards demonstrating the strength of the pi-FISH method, specifically with respect to sensitivity. We still however have some concerns that need to be addressed in a revision. Once addressed, we believe the method will represent an important methodological advance for spatial omics technology and (with the more quantitative assessment requested below) would be beneficial for the overall community. Notably, our comments should be addressable solely by analysis of data already included in the manuscript.

1. The data in the revised manuscript convincingly show the detection of lowly expressed genes, but we still believe the authors could provide better quantitative guidance on the ability of pi-FISH to reliably detect signal vs. noise. We're specifically concerned with understanding where the noise limit (floor) of pi-FISH is, so it can be reliably used in quantitative and highly multiplexed methods. Specifically, we'd like to understand how much non-specific binding can be expected in both tissue culture and when applied to tissue sections.

These questions can and should be resolved by further quantitative analysis of pre-existing data present in this revision, as follows:

- In Supplementary Fig. 5e the authors show a 0.36% noise level for a non-specific probe relative to ACTB in tissue culture. How many false positive puncta per cell does this translate to? What is the cell to cell variance in false positive binding?

- In Fig. 4 the authors show very low levels of background noise for the pi-FISH probes in tissue (seemingly lower than cell culture, which is surprising). Could this be replotted with an axis range of 0-10 spot counts/cell (instead of the current 0-100) to reveal if any background noise is present? Are background rates of the dapB probe similar to cell culture data? If so, shouldn't we

expect to see at least 50 background counts of dapB in Supplementary Fig. 4e?

- Can the authors please add a plot where the mutually exclusive gene expression data in Supplementary Fig. 6e,f are also presented as a scatter plot of gene expression per cell to demonstrate quantitative mutual exclusion?

- Fig. R2-5 would benefit from a scatter plot of single molecule HCR vs. pi-FISH intensity per FISH foci so we can quantitatively assess the correlation between the two methods

2. The analysis of MERFISH vs Pi-FISH rainbow convincingly shows similar regionalization of many of the genes displayed. This analysis should however be further improved by dissecting the data at a single cell level to show if genes shared between the previously published data and pi-FISH rainbow show similar correlations. From the regionalization data presented we expect that they do, but it would be much more convincing to show these data directly.

Reviewer #3:

Remarks to the Author:

The authors have made significant improvements to the manuscript, such as the inclusion of a more detailed methods section, as well as necessary experimental controls. Targeting various loci with minimal probes is a desired goal of the spatial-omics field, which will help enable detection of splice variants which the authors note.

1. The authors claim that triple detection of DNA, RNA and protein has not previously been realized is not accurate. See "Integrated spatial genomics reveals global architecture of single nuclei (2021), Takei et al, Nature." I recommend tempering this assertion in the manuscript.

2. The cytoplasm is extremely crowded with the 25 pi-fish probe condition in Figure R3-1, as well as in other experiments- Fig R3-6. In general, RNA spot detection in crowded cellular environments is a current challenge in the field of spatial-omics. Thus, it is difficult to appreciate the accuracy of the RNA spot detection and fluorescence intensity measurements in this figure using the described imaging analysis pipeline.

Reviewers' comments:

Reviewer #1 (Remarks to the Author):

The authors have added a substantial amount of control data and addressed the reviewers' comments in the revised manuscript. The addition of schematics and clarifications in the methods should help other laboratories implement the π -FISH and π -FISH+ methods. The description of π -FISH/ π -FISH+ as new in situ detection methods should be valuable to laboratories that may not have the ability to use other higher throughput in situ analysis methods. The increased sensitivity should be useful for detection of rare or small RNAs, short exons, and sequence variants.

Response: Thank you for your positive comments and valuable suggestions. We have made revisions accordingly. All changes have been tracked with yellow highlights in the revised manuscript. Please also refer to our point-by-point responses below.

Specific comments:

Fig. S2. FWHM is not defined in legend (full width at half maximum?).

Response: Thank you for the comment. As suggested, we have added the definition of FWHM (full width at half maximum) in the legend of Supplementary Fig. 2h in the revised manuscript.

Fig. S3g, h: why is multichannel detection more efficient than single channel for both probes? If multichannel spots must be detected with all colors to be scored, they should be less frequent than the single channel detection frequencies. If multichannel spots are being scored as positive with missing channels (e.g., 2 of 3 channels), that could lead to a higher percentage of detection, but it would be misleading since a multichannel spot with one color missing could lead to incorrect probe identification when multiple targets are detected together.

Response: We agree with your comment that multi-channel detection should be less efficient than single-channel. Indeed, we used the calculation method as you suggested. However, the way we presented our data in Supplementary Fig. 3g and h is not clear and may be confusing. As shown in Fig. R1-1d and e, our data indicated that the multi-channel detection efficiency was indeed slightly lower than the single-channel detection efficiency. As described in the manuscript, "The detection efficiencies of *Gad1* and *Vip* genes by multi-channel decoding were 99.03% and 99.10% of single-channel detection, respectively." (lines 154–156 in the revised manuscript). To avoid

confusion, we updated Supplementary Fig. 3g and h to unambiguously present the data.

Fig. R1-1. Efficiency of π -FISH rainbow in multiplexed detection.

(a-b) *In situ* detection of *Gad1* (a) and *Vip* (b) genes expression with triple channels in mouse brain tissues. Scale bars, 100 μ m.

(c) *Gad1* mRNA (left) and *Vip* mRNA (right) were detected with a single channel in mouse brain tissues, respectively. Scale bars, 100 μ m.

(d-e) The efficiency of multichannel detection for *Gad1* (d) and *Vip* (e) from (a-c) were 99.03% and 99.10%, in comparison to single-channel detection, respectively. n = 30 cells.

Fig. S16c. The additional examples of miR-145-5p detection, as well as the additional controls are helpful. The correlation between miR-145-5p and Malat1 subcellular localization is not as apparent as in Fig. 5b (other than that both are in the nucleus), but the experiment still demonstrates the ability to detect overlapping expression between miRNAs and lncRNAs in cells.

Response: Thank you for the comment. During co-detection of *MALAT1* and *miR145-*

5p by π -FISH+, we observed two expression sub-patterns of *MALAT1*: a proportion of cells showed a more diffuse distribution of *MALAT1* and *miR145-5p* in the nucleus, while others showed larger aggregated *MALAT1* and *miR145-5p* signals. Interestingly, aggregated *MALAT1* showed a more striking colocalization pattern with *miR145-5p*. Thus, we only showed the colocalization of *miR145-5p* with the aggregated *MALAT1* signals in Fig. 5b in the previous manuscript. It would be of great interest to further investigate the dynamic transition between different subcellular distributions of *MALAT1* and the associated functions, especially in conjunction with *miR145-5p*. These data support the ability of π -FISH+ for simultaneous detection of LncRNA and miRNA. As suggested, we added more data on the two sub-patterns of the colocalization of *MALAT1* and *miR145-5p* in Supplementary Fig. 16c in the revised manuscript (Fig. R1-2a here).

Moreover, as suggested, we performed several controls for *miR145-5p* detection, including a scramble extension probe of *miR145-5p* and no extension probe, to further verify the specificity of *miR145-5p* detection. As shown in Fig. R1-2b, signals were barely observed in the negative control groups, further confirming the specificity of *miR145-5p* detection.

Fig. R1-2. Co-detection of *miR145-5p* and *MALAT1* by π -FISH+.

(a) The *miR145-5p* and its sponge LncRNA *MALAT1* were co-detected by π -FISH+. Two sub-patterns: some cells show a more diffuse distribution of both signals in the nucleus, while others show larger aggregated signals. Scale bars, 2.5 μ m.

(b) The specificity of the *miR145-5p* complementary sequence within the extension probe was verified with a scramble extension probe (nonspecific binding) and no extension probe for the hybridization. Scale bars, 2.5 μ m.

Fig. S17 may be easier for readers to understand if the figure legend states the conclusions from the figure: why is detection of the two ARV7 loci of interest (currently only stated in the main text)?

Response: As suggested, we have added information to the legend of Supplementary Fig. 17 in the revised manuscript as follows:

ARV7 was identified by simultaneously labelling *ARV7* locus 1 (transcript in *ARV1*, *ARV2*, *ARV4*, and *ARV7*) and locus 2 (transcript in *ARV5* and *ARV7*) using two

fluorescence signal probes. The merged signal (yellow) of locus 1 (magenta) and locus 2 (red) represents *ARV7* signals as well as the *K19* signal (green) as a CTCs biomarker.

Reviewer #2 (Remarks to the Author)

We thank the authors for their detailed revisions which go a long way towards demonstrating the strength of the pi-FISH method, specifically with respect to sensitivity. We still however have some concerns that need to be addressed in a revision. Once addressed, we believe the method will represent an important methodological advance for spatial omics technology and (with the more quantitative assessment requested below) would be beneficial for the overall community. Notably, our comments should be addressable solely by analysis of data already included in the manuscript.

Response: Thank you for your comments and valuable suggestions. As suggested, we have provided further quantification analysis for the experiments presented in the manuscript. All changes have been tracked with yellow highlights in the revised manuscript. Please also refer to our point-by-point responses below.

1. The data in the revised manuscript convincingly show the detection of lowly expressed genes, but we still believe the authors could provide better quantitative guidance on the ability of pi-FISH to reliably detect signal vs. noise. We're specifically concerned with understanding where the noise limit (flood) of pi-FISH is, so it can be reliably used in quantitative and highly multiplexed methods. Specifically, we'd like to understand how much non-specific binding can be expected in both tissue culture and when applied to tissue sections.

Response: Thank you for the comment. As suggested, we clearly show the number of nonspecific signals in cell culture and tissue sections, as per the following responses.

These questions can and should be resolved by further quantitative analysis of pre-existing data present in this revision, as follows:

- In Supplementary Fig. 5e the authors show a 0.36% noise level for a non-specific probe relative to ACTB in tissue culture. How many false positive puncta per cell does this translate to? What is the cell to cell variance in false positive binding?

Response: Thank you for your valuable comment. As suggested, we have added quantitative scatter plots to show false positive puncta per cell and the cell-to-cell variance in false positive binding. As shown in Fig. R2-1e-h, the number of false positive signals per cell in single and multiplex detection was extremely low compared with positive signals, suggesting high specificity and low background noise of π -FISH rainbow. We have added these quantitative data as Supplementary Fig. 5i-l in the

Fig. R2-1. Verification of the false positive rates in single and multiplexed detection of π -FISH rainbow.

(a-b) False positive rates of single (e) and multiplexed (f) detection of π -FISH rainbow were verified by detecting *ACTB* expression alone and simultaneously detecting *ACTB*, *PPIA*, and *B2M* expression, respectively, along with nonspecific probes (bacterial *dapB* probes) addition. The false positive rate of single detection of π -FISH rainbow is the ratio of the total false positive spot counts of *dapB* (green) over the total spot counts of *ACTB*. The false positive rate of multiplexed detection of π -FISH rainbow is the ratio of the false positive spot counts (total spot counts of *ACTB* (red), *PPIA* (green), *B2M* (magenta), and *dapB* (encoded by three channels) minus the total spot counts of *ACTB*, *PPIA*, and *B2M*) over the total spot counts of *ACTB*, *PPIA*, and *B2M*. Scale bars, 10 μ m.

(c-d) The false positive rate of single (g) and multiplexed detection (h) of π -FISH rainbow were further verified by detecting the *PPIA* gene expression alone and simultaneously detecting *ACTB*, *PPIA*, and *B2M* genes expression, with bilaterally specific and unilaterally specific π target probes. The false positive rate for *PPIA* detection by π -FISH rainbow is the total spot counts of *PPIA* with unilaterally specific π target probes over the total spot counts of *PPIA* with bilaterally specific π target probes. The false positive rate of multiplexed detection of π -FISH rainbow is the total spot counts of *ACTB* (red), *PPIA* (green), and *B2M* (magenta) with unilaterally specific π target probes

over the total spot counts of *ACTB*, *PPIA*, and *B2M* with bilaterally specific π target probes. Scale bars, 10 μm .

White numbers in the images of (e) to (h) indicated false positive rates. $n = 50$ cells per group.

(e-h) The histogram of spot counts per cell for positive signals and background noise from (a-d).

- In Fig. 4 the authors show very low levels of background noise for the pi-FISH probes in tissue (seemingly lower than cell culture, which is surprising). Could this be replotted with an axis range of 0-10 spot counts/cell (instead of the current 0-100) to reveal if any background noise is present? Are background rates of the *dapB* probe similar to cell culture data? If so, shouldn't we expect to see at least 50 background counts of *dapB* in Supplementary Fig. 4e?

Response: Thank you for the comment. Our data showed that the noise per cell ranged from 0 to 5 and 0 to 6 counts for cell and tissue detection, respectively. As suggested, we redrew the scatter plot with an axis range of 0-10 spot counts per cell and increased the sample size to 60 cells to display the amount of background noise in each cell. As shown in Fig. R2-2e-h, the background rates of the *dapB* probe in cells and tissues were comparable, indicating the high specificity and low background noise of π -FISH rainbow. We have added these figures as Supplementary Fig. 4 in the revised manuscript.

Fig. R2-2. Background of π -FISH rainbow in cells and tissues.

(a-b) The background of π -FISH rainbow in cells was verified by *in situ* detection of *ACTB* (a) and co-detection of *ACTB*, *PPIA*, and *B2M* (b) along with multiple negative controls, including unilateral target probes, no target probes, bilateral target probes with RNase treatment, and probes targeting the bacterial gene *dapB* in HeLa cells, respectively. Scale bars, 10 μ m.

(c-d) The background of π -FISH rainbow in tissues was verified by *in situ* detection of *Ctgf* (c) and co-detection of *Ctgf*, *Penk*, and *Sst* (d) along with multiple negative controls, including unilateral target probes, no target probe, bilateral target probes with RNase treatment, and probes targeting the bacterial gene *dapB* in mouse brain sections, respectively. (i-v) Higher magnification of square regions in the upper panels of (c) and (d) were shown in the lower panels. Scale bars, 100 μ m (top) and 20 μ m (bottom).

(e-h) The histogram of spot counts per cell for positive signals and background noise from (a-d). n = 60 cells per group.

- Can the authors please add a plot where the mutually exclusive gene expression data in Supplementary Fig. 6e,f are also presented as a scatter plot of gene expression per cell to demonstrate quantitative mutual exclusion?

Response: As suggested, we have added quantitative analyses of mutually exclusive gene expressions as scatter plots. As shown in Fig. R2-3, quantitative mutual exclusion was clearly observed in the mutually exclusive genes *Slc17a7* and *Gad1*, as expected. There was almost no *Gad1* expression (green) in *Slc17a7*⁺ neurons (red). Likewise, there was scarcely any *Slc17a7* expression in *Gad1*⁺ neurons. We also observed the same mutually exclusive expression pattern for *Vip* and *Sst*. These results demonstrate the specificity and accuracy of π -FISH rainbow. We have added these results as Supplementary Fig. 7e and f in the revised manuscript.

Fig. R2-3. The specificity of π -FISH rainbow was verified by detecting the mutually exclusive

expression of different neuron markers.

(a) Co-detection of *Sst* and *Vip*, two mutually exclusively expressed genes in different subclasses of interneurons in the mouse cerebral cortex, by π -FISH rainbow. The green and red arrows indicated *Sst*⁺ and *Vip*⁺ neurons, respectively. Scale bars, 100 μ m.

(b) The scatter plot of spot counts per cell for mutually exclusive genes *Vip* and *Sst* from (a). n = 89 cells.

(c) Co-detection of inhibitory (*Gad1*) and excitatory (*Slc17a7*) neuron markers in mouse cerebral cortex by π -FISH rainbow. The green and red arrows indicated *Gad1*⁺ and *Slc17a7*⁺ neurons, respectively. Scale bars, 100 μ m.

(d) The scatter plot of spot counts per cell for mutually exclusive genes *Slc17a7* and *Gad1* from (c). n = 848 cells.

- Fig. R2-5 would benefit from a scatter plot of single molecule HCR vs. pi-FISH intensity per FISH foci so we can quantitatively assess the correlation between the two methods.

Response: Thank you for this comment. As suggested, we have measured the fluorescence intensity of individual spots and added scatter plots to assess the correlation between π -FISH rainbow and HCR. As shown in Fig. R2-4, our data showed that the fluorescence intensity of individual spots was highly correlated between the two methods.

Fig. R2-4. Detection of *Ctgf* and *Gad1* by both π -FISH rainbow and HCR.

(a and c) *Ctgf* (a) and *Gad1* (c) were respectively detected in the mouse brain tissues by both π -FISH rainbow and HCR. White boxes represented the higher-resolution images. The red and green arrows indicated π -FISH rainbow and HCR signals, respectively. Scale bars, 100 μ m.

(b and d) Correlation of fluorescence intensity of signal spots between π -FISH rainbow and HCR

from (a) and (c), respectively. r stands for the Pearson correlation coefficient.

2. The analysis of MERFISH vs Pi-FISH rainbow convincingly shows similar regionalization of many of the genes displayed. This analysis should however be further improved by dissecting the data at a single cell level to show if genes shared between the previously published data and pi-FISH rainbow show similar correlations. From the regionalization data presented we expect that they do, but it would be much more convincing to show these data directly.

Response: As suggested, we have added correlation analyses of genes shared by π -FISH rainbow and MERFISH at the single-cell level. As shown in Fig. R2-5, the RNA copy number of the *Rorb*, *Cux2*, *Sulf2*, *Foxp2*, and *Cxcl14* genes per cell detected by π -FISH rainbow are compared to that by MERFISH. The measurement results of the two methods were highly correlated, further indicating the reliable performance of π -FISH rainbow.

Fig. R2-5. Correlation between π -FISH rainbow and MERFISH data.

(a-e) Correlation analyses of signal spot counts of genes (*Rorb*, *Cux2*, *Sulf2*, *Foxp2*, and *Cxcl14*) shared by π -FISH rainbow and MERFISH at the single-cell level.

Reviewer #3 (Remarks to the Author):

The authors have made significant improvements to the manuscript, such as the inclusion of a more detailed methods section, as well as necessary experimental controls. Targeting various loci with minimal probes is a desired goal of the spatial-omics field, which will help enable detection of splice variants which the authors note.

Response: Thank you for your positive feedback and valuable suggestions, which significantly improved our manuscript. Please refer to our point-by-point responses to your comments below.

1. The authors claim that triple detection of DNA, RNA and protein has not previously been realized is not accurate. See “Integrated spatial genomics reveals global architecture of single nuclei (2021), Takei et al, Nature.” I recommend tempering this assertion in the manuscript.

Response: Thank you for this comment. As suggested, we have toned down the assertions in the introduction and discussion part of the revised manuscript to make our claim more accurate.

In the introduction of the revised manuscript, we revised the text to “Several excellent studies using MERFISH and seqFISH+ methods have co-detected DNA, RNA, and proteins in the same sample^{1,2}. The co-detection of three molecules by these methods was achieved through separated hybridization, imaging, and subsequent integrated signal analysis, which is more time-consuming and challenging, especially for the alignment of 3D images. Thus, a more efficient and accurate one-step simultaneous triple detection method for DNA, RNA, and proteins is highly desired”.

We further revised the text in the discussion of the revised manuscript “Although many FISH methods can detect DNA, RNA, and proteins individually, there are no methods to simultaneously detect genomic loci, RNA transcripts, and proteins” to “Although many FISH methods can co-detect DNA, RNA, and proteins^{1,2}, these methods rely on multiple rounds of hybridization, imaging, and alignment. Thus, an efficient and accurate simultaneous detection method of genomic loci, RNA transcripts, and proteins in one-step imaging is desperately needed”.

2. The cytoplasm is extremely crowded with the 25 pi-fish probe condition in Figure R3-1, as well as in other experiments- Fig R3-6. In general, RNA spot detection in crowded cellular environments is a current challenge in the field of spatial-omics. Thus, it is difficult to appreciate the accuracy of the RNA spot detection and fluorescence

intensity measurements in this figure using the described imaging analysis pipeline.

Response: Thank you for this comment. We agree that the crowding environment of RNA signal spots has always been a challenge for spatial omics. π -FISH rainbow was designed for low to medium throughput detection (one round of hybridization and imaging detects ≤ 15 genes), reducing the challenge of signal decoding. Moreover, π -FISH rainbow can divide several highly expressed genes into multiple rounds for multiplexed detection, further reducing signal decoding challenges. For the detection of highly expressing genes like *PPIA* by π -FISH rainbow: while the signals looked crowded in the small figures in our study, the signal dots could indeed be differentiated and recognized by the Imaris software (version 9.9), as shown in Fig. R3-1.

Fig. R3-1. Detection of *PPIA* mRNA using different numbers of π target probes by π -FISH rainbow.

(a-d) Detection of *PPIA* mRNA in HeLa cells with 10, 15, 20, and 25 pairs of π target probes, respectively. Higher magnification of yellow square regions in the upper panels were shown in the lower panels. Red dots represented the identified signal spots. Each spot could indeed be differentiated and recognized by the Imaris software (version 9.9). Scale bars, 10 μm (up) and 5 μm (middle and bottom).

References:

1. Takei Y, Yun J, Zheng S, Ollikainen N, Pierson N, White J, *et al.* Integrated spatial genomics reveals global architecture of single nuclei. *Nature* 2021, **590**(7845): 344-350.
2. Su JH, Zheng P, Kinrot SS, Bintu B, Zhuang X. Genome-Scale Imaging of the 3D Organization and Transcriptional Activity of Chromatin. *Cell* 2020, **182**(6): 1641-1659 e1626.

Reviewers' Comments:

Reviewer #1:

Remarks to the Author:

In their latest revision, the authors have addressed the issues raised by the reviewers. I think the work will be of interest to a wide range of laboratories.

Reviewer #2:

Remarks to the Author:

The modifications to the figures the authors made go a long way to improving the quality of the manuscript. We recommend acceptance in the current form and congratulate the authors on their method and study.

Reviewers' comments:

Reviewer #1 (Remarks to the Author):

In their latest revision, the authors have addressed the issues raised by the reviewers. I think the work will be of interest to a wide range of laboratories.

Response: We sincerely thank you for your positive comments and valuable suggestions during the review process, which are greatly helpful in improving the quality of the manuscript.

Reviewer #2 (Remarks to the Author)

The modifications to the figures the authors made go a long way to improving the quality of the manuscript. We recommend acceptance in the current form and congratulate the authors on their method and study.

Response: We really appreciate your inspiring suggestions and comments, which have significantly improved our work.